# Data-driven Staircase Activation Functions for Ordinal Classification

## Abstract

Ordinal labels are discrete and ordered but lack calibrated spacing, a structure that most deep networks ignore by treating them as nominal classes or real values. We introduce trainable staircase activations as a drop-in replacement, which partitions the output space into learnable, ordered intervals to align predictions with the ordinal labels. Direct parameterization reveals a degeneration–saturation dilemma in which gradients vanish and intervals collapse; we analyze its cause and propose three remedies: (i) stochastic noise injection to de-saturate plateaus, (ii) a monotonic ascending term to enforce order, and (iii) adaptive piecewise-linear functions that adjust thresholds end-to-end. Paired with a mutual information regularized absolute-error loss, our design stabilizes optimization and preserves ordinal structure. The modules are drop-in replacements for final layers and integrate with standard architectures without any architectural changes. Across diverse benchmarks, they consistently outperform softmax/logistic baselines and prior ordinal methods, demonstrating that staircase activations are an effective and principled building block for end-to-end learning with ordinal targets.

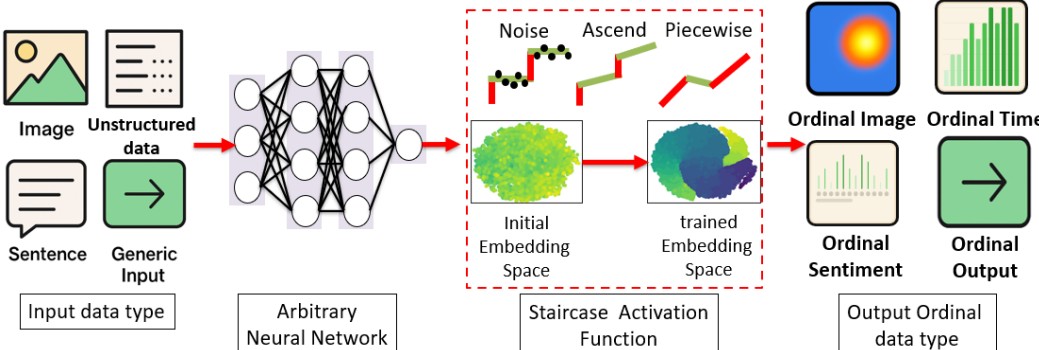

Figure 1: End-to-end ordinal prediction: inputs (images, unstructured data, text, etc.) are processed by a chosen model (e.g., MLP, ResNet, BERT, GPT, Transformer). The proposed Staircase Activation, Noise, Ascending, or Piecewise, maps outputs into ordinal predictions across domains such as vision, time series, and sentiment.

## 1    Introduction

Ordinal labels (e.g., AAA–BB in credit ratings Huang et al. (2004)) denote discrete but ordered categories and appear across domains Huang et al. (2004); Kyriazopoulou et al. (2021); Bürkner & Vuorre (2019); Georgoulas et al. (2016). Classical methods, such as regressors, SVMs, trees, probabilistic kernels, and ensembles, often reduce the task to multiple binary classification tasks Gutiérrez et al. (2015); Cardoso & da Costa (2007). Many deep models still treat ordinal prediction as regression or nominal classification, lacking order-aware activations Vargas et al. (2023); Zhang et al. (2023); Cardoso et al. (2025). Label encodings avoid architectural changes and thus remain popular: NNRank converts one-hot to unary to preserve order Cheng et al. (2008), while BEL codes transform one-hot into order-aware strings Shah et al. (2022). Beyond encodings, simple baselines

exist, e.g., GLPDepth scales a sigmoid output to represent levels Kim et al. (2022), and MORF frames ordinality via tree decisions in medical imaging Lei et al. (2022). Loss-/label-driven approaches (e.g., CORAL, OLL) enforce order externally Cao et al. (2020); Castagnos et al. (2022) leaving internal representations largely order-agnostic.

Despite the growing interest in task-specific activations, ordinal-aware functions remain underexplored Ramachandran et al. (2018); Gao et al. (2023); Zhu et al. (2023). Staircase-like activations align naturally with ordered labels by partitioning outputs into ordered intervals but face *saturation* and non-differentiability, hindering gradient-based learning Tóth & Gosztolya (2004); Koçak & Şiray (2021). Prior reports show that literal staircase functions fail to converge under backpropagation, whereas ramp-like or sigmoid activations train successfully Tóth & Gosztolya (2004); step-like variants (multi-sigmoid Boskovitz & Guterman (2002), MSAF Koçak & Şiray (2021), and other multilevel designs Piraud et al. (2018); Hu et al. (2019); Konar et al. (2022)) require delicate threshold/steepness tuning and have seen limited uptake, with no clear gains over ReLU in general-purpose practice.

We introduce trainable *staircase* activations that generalize binary steps to multi-level, learnable intervals for ordinal prediction and explicitly expose a degeneration–saturation dilemma. We then propose three complementary remedies—(i) stochastic noise injection, (ii) a monotonic ascending constraint, and (iii) a monotone piecewise-linear parameterization—paired with a Mutual-Information (MI)–regularized MAE loss, motivated by evidence that recasting regression as classification can increase mutual information (MI) Gu et al. (2022); Zhang et al. (2023). Only replacing the output layer in public baselines yields consistent gains while preserving the ordinal structure end-to-end. Our contributions are:

- **Staircase activations.** A trainable, multi-level staircase that preserves order and reveals the degeneration–saturation dilemma.
- **Theory and remedies.** Analysis plus three fixes—noise, ascending constraint, and piecewise-linear parameterization—together with an MI-regularized loss that stabilizes optimization.
- **Experiments.** Across diverse applications and strong baselines, our modules deliver consistent gains and enable end-to-end extraction of ordinal structure.

Further related work appears in Appendix C.

## 2 BACKGROUND AND MOTIVATIONS

This section introduces the staircase functions and the challenges they face within the backpropagation mechanism. Subsequently, theoretical analyses via gradient and mutual information are provided.

### 2.1 BINARY STEP ACTIVATION FUNCTIONS

The Soft-Sigmoid (ss) and Soft-Tanh (st) functions, derived from the Sigmoid and Tanh (Eq. 1) functions, focus on smooth transitions, while the Hard-Sigmoid (hs) and Hard-Tanh (ht)(Eq. 2) were specifically proposed to facilitate noise injection Nwankpa et al. (2021).

$$\phi^{ss}(z, T) = \frac{1}{1 + e^{-Tz}} = \text{Sigmoid}(z, T), \qquad \phi^{st}(z, T) = \frac{e^{Tz} - e^{-Tz}}{e^{Tz} + e^{-Tz}} = \text{Tanh}(z, T) \qquad (1)$$

$$\phi^{hs}(z, T) = \max\{\min\{u^s(Tz), 1\}, 0\}, \qquad \phi^{ht}(z, T) = \max\{\min\{u^t(Tz), 1\}, -1\} \qquad (2)$$

Let $z$ denote the latent vector from the preceding layer, and let $T$ be the logistic shape (temperature) parameter. We write the temperature-scaled logistic as $\sigma_T(x) := \sigma(Tx)$, where increasing $T$ steepens the curve and decreasing $T$ flattens it Kokkinis et al. (2011). Its influence is illustrated in Appendix A, Fig. 1. For local linear references around $x = 0$, we use first-order Taylor expansions: $u^s(x) \triangleq 0.25\,x + 0.5$ for sigmoid and $u^t(x) \triangleq x$ for tanh. Sigmoid and tanh act as smooth relaxations of the Heaviside step function, replacing the discontinuous jump with a differentiable transition that is amenable to gradient-based optimization.

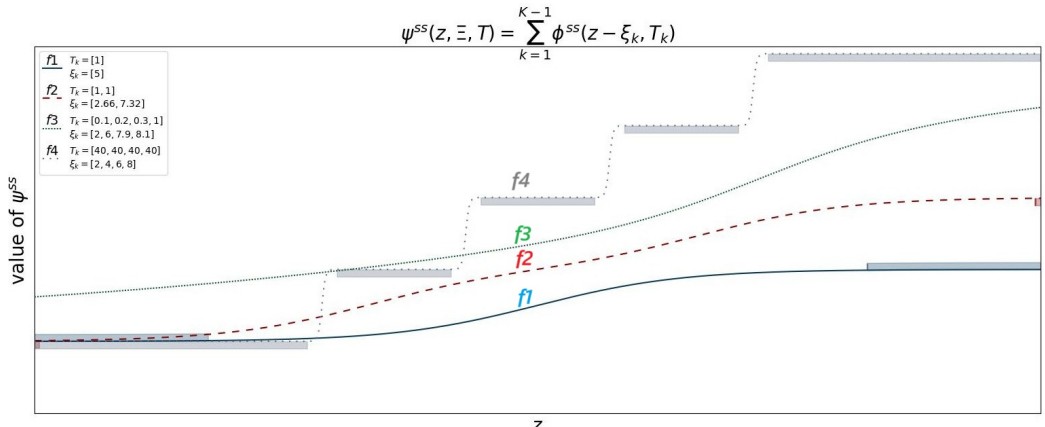

Figure 2: Shaded bands mark saturation ($gradient < 10^{-2}$). $f_1$ shows two stages (center-shifted sigmoid); $f_2$ shows three. $f_3$ and $f_4$ target five stages, yet $f_2$ and $f_3$ degenerates, and as $f_4$ approaches a true five-step staircase, saturation dominates the domain, hindering training. Although $f_4$ saturates, $f_2$ and $f_3$ avoid saturation yet degenerate, preventing reliable ordinal learning.

## 2.2 STAIRCASE ACTIVATION FUNCTIONS

Assume inputs $x \in \mathcal{X}$ carry ordinal structure, and there exists an activation $f : \mathcal{X} \to \mathbb{R}$ that induces $K$ ordered labels. Then $f$ should satisfy two properties: (i) *monotonicity* to preserve order, $\forall\, x_i < x_j :\; f(x_i) \le f(x_j)$; and (ii) *within-class closeness versus across-class separation*: if $x_i, x_j$ belong to the same class while $x_k$ belongs to a different class, then

$$|f(x_i) - f(x_j)| \;<\; \min\big\{|f(x_k) - f(x_i)|,\; |f(x_k) - f(x_j)|\big\}.$$

For example, with $x_i{=}2$, $x_j{=}3$ (same class) and $x_k{=}10$ (different class), if $f(x_i){=}0.2$, $f(x_j){=}0.3$, $f(x_k){=}1.0$, we have $|0.2{-}0.3|{=}0.1 < \min\{0.8, 0.7\}$.

Extending the Heaviside step (and its soft/hard approximations; Appendix A Fig. 1(a)) from binary classification to a $K$-level *staircase* satisfies these desiderata. In the binary case, Appendix Fig. 1(a) corresponds to a single threshold at $z{=}0.5$. The temperature $T$ controls the transition steepness—shrinking the central transition band while enlarging the saturation zones away from the threshold—whereas the thresholds $\Xi = \{\xi_1, \ldots, \xi_{K-1}\}$ determine the jump locations. Motivated by these observations, we study $K$-level staircase activations parameterized by $\{T, \Xi\}$.

For $K > 2$, closed-form staircases can be composed from binary steps (Eqs. 1, 2) Koçak & Şiray (2021). We generalize them to a learnable family with thresholds $\Xi = \{\xi_k\}_{k \in [K-1]}$ and per-level shape parameters $T = \{T_k\}_{k \in [K-1]}$ tailored to ordinal outputs.

**Definition 1** (Staircase Activation Function). *Given thresholds $\Xi = \{\xi_1, \ldots, \xi_{K-1}\}$ and shape parameters $T = \{T_1, \ldots, T_{K-1}\}$ with $\xi_k < \xi_{k+1}$, the multi-level staircase activations are*

$$\psi_{\Xi,T}^{ss,hs}(z) = \sum_{k=1}^{K-1} \phi^{ss}(z - \xi_k, T_k), \qquad \psi_{\Xi,T}^{st,ht}(z) = \sum_{k=1}^{K-1} \frac{\phi^{st}(z - \xi_k, T_k) + 1}{2} \qquad (3)$$

*where $\phi^{ss}, \phi^{st}$ are the soft (sigmoid/tanh) bases and $\phi^{hs}, \phi^{ht}$ are their hard counterparts.*

**Sigmoid-only focus.** Since $\tanh(z) = 2\,\sigma(2z) - 1$, we restrict attention to *sigmoid*-based staircases; by symmetry we assume $T_k > 0$ for all $k \in [K - 1]$. Detailed provided in Appendix D.

**Parameters and notation.** The thresholds $\Xi = \{\xi_1, \ldots, \xi_{K-1}\}$ and shape (temperature) parameters $T = \{T_1, \ldots, T_{K-1}\}$ can be fixed or learned (Sec. 3). To ensure the logit range is covered, introduce dummy bounds $\xi_0 < \xi_1 < \cdots < \xi_{K-1} < \xi_K$ with $z \in (\xi_0, \xi_K)$. For brevity we write $\psi^{ss}$ (soft-sigmoid staircase) or $\psi^{hs}$ (hard-sigmoid staircase). These are staircase-*like* (no vertical corners at $\Xi$), but we simply call them "staircase" henceforth.

**Challenges with parameterized staircases.** Fig. 2 shows four staircase activations under different $(\Xi, T)$ (we provided seven staircase activation version in Appendix E). $f_1$ is the standard binary ($K{=}2$) sigmoid; $f_2$ is a 3-state MSAF Koçak & Şiray (2021). For $K{=}5$, $f_3$ *degenerate*: the shape fails to realize five distinct levels, revealing a tight coupling between thresholds and shapes. Only $f_4$ realizes $K{=}5$, yet—unlike $f_1$ whose saturation lies far from the transition—its saturation bands concentrate within the main $z$-range (Shaded bands intervals). This underscores the need for principled parameterization/learning to avoid degeneration and in-band saturation.

## 2.3 DEGENERATION-SATURATION DILEMMA

This subsection elucidates the aforementioned phenomena observed in the context of parameterized staircase functions. The term **degeneration** of a staircase-like function refers to a scenario in which the number of states presented by its function plot is lower than the number of categories corresponding to its configuration in ordinal classification task. The **saturation** of sigmoid and softmax functions Gulcehre et al. (2016); Chen et al. (2017) highlights that ineffective training occurs due to the gradient approaching zero as $z \to \pm\infty$. This concept can be directly extended to parameterized staircase-like functions. Empirical observations suggest that insufficiently large values of $T_k$ can lead to various forms of degeneration. Following the Taylor approximation, as an example to mitigate the risk of severe degeneration, it is advisable to set $T_k$ satisfying

$$T_k > 10 \times \max\{(\xi_k - \xi_{k-1})^{-1}, (\xi_{k+1} - \xi_k)^{-1}, 1\} \tag{4}$$

for each pair of thresholds.

**Definition 2.** *(Degeneration-Saturation Dilemma)*
*For the function $\psi_{\Xi,T}^{ss,hs}(z)$ with given $\{\Xi, T\}$, if $\forall k \in [K-1]$, Eq.(4) holds, then $\psi_{\Xi,T}^{ss,hs}(z)$ can circumvent degeneration but may encounter saturation. On the other hand, if $\exists k \in [K-1]$ s.t. Eq.(4) does not hold, then degeneration occurs for $\psi_{\Xi,T}^{ss,hs}(z)$, but saturation is likely to be avoided for the corresponding $z \in (\xi_{k-1}, \xi_{k+1})$.*

Observe that degeneration and saturation can occur in a function with a bad setting of $\{\Xi, T\}$, such as $f_2$ and $f_3$ in Fig. 2. To perform a detailed analysis of saturated behavior, we shift our attention to the domain of the activation function.

**Definition 3.** *($\frac{1}{\epsilon}$-Near Saturation domain)*
*For a given Staircase activation function $\psi : \mathbb{R} \to \mathbb{R}$ and a small number $0 < \epsilon \ll 1$, the $\frac{1}{\epsilon}$ near saturation domain is $\mathcal{SD}_{\psi,\epsilon} \triangleq \left\{ z \in \mathbb{R} : |\frac{d\psi(z)}{dz}| \leq \epsilon \right\}$. If $\epsilon \to 0$ then $\mathcal{SD}_{\psi,0}$ is the perfect (infinitely-near) saturation domain.*

We call the thresholds and shape parameter set $\{\Xi, T\}$ **a uniform set** if $\forall k \in [K]$, $\xi_{k+1} - \xi_k = \xi_k - \xi_{k-1}$, and $T_k = T_{k-1}$. Two activation functions, say $\psi_i(z)$ and $\psi_j(z)$, are **equivalent** in the backpropagation mechanism if there exist positive constants $c_1, c_2$ such that $c_1 \frac{d}{dz}\psi_i \leq \frac{d}{dz}\psi_j \leq c_2 \frac{d}{dz}\psi_i$. The following proposition suggests that the non-uniform setting of thresholds, similar to the discussion in Zhu et al. (2023), and that of shape, are crucial characteristics for the staircase function to demonstrate an advantage over ReLU or hard-sigmoid.

**Proposition 1.** *If the staircase-like function $\psi$ has uniformly set parameters and is without the $10^2$-near saturation domain for $z \in (\xi_1, \xi_K)$, then it is equivalent to the hard-sigmoid Eq.(2). Furthermore, if the activation function has no $10^2$-near saturation domain in the open interval $(\xi_1, \infty)$, then it is equivalent to ReLU.*

The proof of this proposition is straightforward by definition and it explains why the third-order staircase-like function proposed in previous works Koçak & Şiray (2021), as $f_2$ as shown in Fig 2, has not received widespread attention. At the same time, it also implies that adaptive or non-uniformly set parameters are necessary.

**Lemma 1.** *Consider the Staircase activation function $\psi$ serving as the output layer of a given deep learning model $F_\Theta$ and having a non-zero length $\frac{1}{\epsilon^2}$-near saturation domain. Let $\gamma$ denote the learning rate. For a training sample $(x, y)$ that $\hat{y}{=}F_\Theta(x) = \psi^{ss}(z)$, and $|\hat{y} - y| \geq 1$,where $y$ is the ordinal label, the training process based on gradient descent requires (at least) $\Omega((\gamma\epsilon)^{-1})$ steps of updating the trainable weights $\Theta$ such that $\hat{y} = y$ if $z(x)$ falls in the $\frac{1}{\epsilon^2}$-near saturation domain $\mathcal{SD}_{\psi,\epsilon^2}$.*

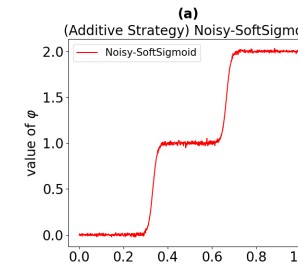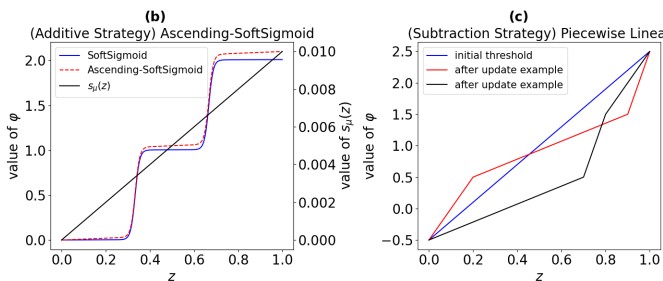

Figure 3: Mitigating saturation (i.e., derivative or gradient less than 1e-4) in Staircase activations. Shown with SoftSigmoid but applicable to all variants: (a) *Additive*—inject a stochastic term (see Appendix. G); (b) *Additive*—add an ascending term $s_\mu(z)$; (c) *Subtraction*—shrink or remove saturated regions via adaptive piecewise linearization.

We defer the complete proof to Appendices E, F, and I. To mitigate the resulting degeneration–saturation dilemma, we can undertake two 'approaches: (1) employ data-driven methods to guide the model in learning appropriate parameter settings, and (2) implement remedial actions when $T_k$ is large.

## 3    METHODS

This section revisits noise injection—a classic fix for saturation in binary activations (e.g., sigmoid). In multi-level *staircase* activations, saturation and *degeneration* worsen, motivating three designs: (i) **Noisy Staircase**, which injects training-time noise to sustain gradients; (ii) **Ascending Staircase**, which adds a monotone slope to guide outputs; and (iii) **Piecewise-Linear Staircase**, which removes flat bands via monotone piecewise interpolation. We study these via gradient dynamics (App. G) and mutual-information preservation (App. L). Each breaks flat, non-differentiable regions to restore backpropagation. Noise and ascending operate only during training and are disabled at test time.

### 3.1    (ADDITIVE STRATEGY) NOISY STAIRCASE ACTIVATION

The saturation phenomenon in activation functions can significantly hinder gradient-based optimization by causing gradients to vanish. Previous works have shown that injecting noise into activations can alleviate this issue in binary scenarios Gulcehre et al. (2016).

Inspired by this, we extend noise injection to multi-level staircase activations, aiming to perturb saturated regions and maintain effective gradient flow during training as follows.

$$\psi_{\Xi,T}^{NO;ss,hs}(z) = \sum_{k=1}^{K-1} \phi_{\triangle,\alpha,\beta,p;T_k}^{noisy}(z - \xi_k) \tag{5}$$

More details can be found in Appendix G, and depicted in Fig. 3(a).

### 3.2    (ADDITIVE STRATEGY) ASCENDING STAIRCASE ACTIVATION

We replace stochastic noise with a deterministic, gently monotone bias by adding a learnable ascending term $s_\mu(z)$ to the staircase activation. This guarantees a nonzero gradient and smoothly guides predictions across ordered levels. As shown in Fig. 3(b), the term reduces saturated flats and stabilizes training, akin to the linear-gradient idea in WGAN Arjovsky et al. (2017). The parameter $\mu$ in Eq. 6 is learned, with $|z^L| \ll \mu$ to keep $s_\mu(z)$ approximately linear over the operating range. We denote this variant by *AS*. Specifically, the ascending term $s_\mu(z) \triangleq sin^{-1}\left(\frac{z}{\mu}\right)$, hence

$$\psi_{\Xi,T}^{AS;ss,hs}(z) = s_\mu(z) + \sum_{k=1}^{K-1} \phi^{ss,hs}(z, \xi_k, T) \tag{6}$$

Based on $|z^L| \ll \mu$ and $\frac{d}{dz^L} sin^{-1}(\frac{z^L}{\mu}) = \frac{1}{\sqrt{\mu^2-(z^L)^2}} \approx \frac{1}{\mu}$, we can derive from the Taylor expansion that $s_\mu(z^L) \approx z^L/\mu$ if $z^L \ll \mu$, indicating it can provide stable progressive gradients.

### 3.3 (SUBTRACTION STRATEGY) PIECEWISE LINEAR ACTIVATION

Unlike additive methods that locally perturb activations, we reshape the activation function to remove saturation. Our piecewise linear reformulation of the staircase activation replaces flat regions with continuous linear segments, ensuring strictly nonzero gradients across the full range and preventing vanishing or degenerate gradients. WLOG, a Piecewise Linear Staircase activation can be defined on $[0, 1]$ (i.e., normalized domain) as follows (full version provided in Appendix B):

$$\psi_{\Xi,T}^{PW}(z) = \frac{\left(z - \frac{\xi_{k+1}+\xi_k}{2}\right)}{\xi_{k+1} - \xi_k} + k, \text{ if } z \in (\xi_k, \xi_{k+1}], k \in \{1, ..., K\} \quad (7)$$

where $\xi_0 \triangleq 0$, $\xi_K \triangleq 1$ and hence $\psi_{\Xi,T}^{PW}(z) = -0.5$ if $-\infty < z \leq 0$ and $\psi_{\Xi,T}^{PW}(z) = K - 0.5$ if $1 < z < \infty$. Note $T_k$ is designed to be $1/(\xi_k - \xi_{k-1})$ in Eq.(7), and the initialization of $\xi_k$ is uniformly located in $[0, 1]$. Then *adaptively*, the learning algorithm decides the new locations of $\xi_k$, under the constraint $\xi_k < \xi_{k+1} \forall k \in [K]$, as illustrated in Fig. 3 (c).

### 3.4 MUTUAL-INFORMATION PERSPECTIVE ON THE STAIRCASE OBJECTIVE

Beyond gradient flow considerations, we establish a theoretical connection between our loss design and mutual information maximization. Under the assumption that features follow a Laplace distribution, we prove that minimizing $L_{MAE}$ with staircase activations is equivalent to minimizing conditional entropy $H(Z|Y)$, which promotes tight clustering of same-class features (Theorem 1, Appendix K).

To complete the mutual information picture $I(Z;Y) = H(Z) - H(Z|Y)$, we introduce a dispersion regularizer: $L_{Div} = -\frac{1}{K-1}\sum_{k=1}^{K-1}|\max(z_k) - \min(z_{k+1})|$ This term encourages larger interclass margins, effectively maximizing feature entropy $H(Z)$ while preserving ordinal structure. The final objective becomes: $L_{MI} = L_{MAE} + \lambda_{Div}L_{Div}$, where $\lambda_{Div}$ denote constant.

This design ensures that staircase activations learn representations where different classes are well-separated ($\max H(Z)$) while maintaining within-class compactness ($\min H(Z|Y)$), naturally aligning with ordinal classification requirements.

## 4 EMPIRICAL STUDY

We evaluate the proposed activation functions across **five domains**—time-series forecasting, age estimation, medical diagnosis, monocular depth estimation, and sentiment analysis—on **eight datasets**: AQI; AFAD and MORPH; Abalone; BUSI; KITTI and NYUv2; and SST-5. The benchmarks span heterogeneous *modalities* and varied *ordinal class counts* (classes number from 3 to 80). ***Our contribution lies in the activation itself. We evaluate it by replacing only the activation in public baselines while keeping the backbone, training protocol, and hyperparameters fixed, thus isolating its effect for fair comparison.*** Full specifications, model backbones, literature baselines, datasets, experimental results, and visualizations are summarized in Appendix M to Appendix R.

**Evaluation.** We evaluate with metrics that respect ordinality: (i) distance-based—MAE, MSE, Accuracy, Precision/Recall/F1; (ii) rank-based—Spearman's $R_s$ and Kendall's $\tau_b$; and (iii) task-specific—CS5 for age estimation and SilogLoss for monocular depth. We first train with $L_{MAE}$ to compare fairly to prior work and isolate the effect of the Staircase activation, then use the full objective $L_{MI} = L_{MAE} + \lambda_{Div}L_{Div}$ with adjusted $\lambda_{Div}$. Reported improvements are relative gains over the corresponding baselines. Metric sets follow task conventions (e.g., MAE/CS5 for age; $\delta$ thresholds and RMSE for depth). In all tables, $\uparrow$ indicates higher is better and $\downarrow$ indicates lower is better.

## 4.1 TIME SERIES FORECASTING TASK

For AQI forecasting, we use a 3-layer MLP (400–350–50) and compare the same backbone against softmax+CE, CORAL Cao et al. (2020), and NNOP Cheng et al. (2008) (softmax+MAE). NNOP is the strongest baseline. Our additive/subtractive staircase variants improve most metrics (Table 1), except precision; replacing $L_{\mathrm{MAE}}$ with $L_{\mathrm{MI}}$ yields further gains. PW-MI is best: recall 0.735 (+3.96% vs. 0.707), F1 0.722 (+4.34% vs. 0.692), accuracy 0.735 (+3.96% vs. 0.707). Full results and metric definitions: Appendices P–Q.

Table 1: Evaluate the proposed method against the best baseline on AQI dataset.

| | AQI (next 8 hour) | | | | | | | |
|---|---|---|---|---|---|---|---|---|
| Model | P ↑ | R ↑ | F1 ↑ | Acc ↑ | MAE ↓ | MSE ↓ | $\tau_b$ ↑ | $R_s$ ↑ |
| NNOP | 0.74 | 0.71 | 0.69 | 0.71 | 0.30 | 0.32 | 0.49 | 0.49 |
| PW-MI | $-1.0\%$ | $+4.0\%$ | $+4.3\%$ | $+4.0\%$ | $-9.6\%$ | $-9.8\%$ | $+4.7\%$ | $+6.8\%$ |

The table shows the improvement percentages relative to NNOPCheng et al. (2008). P and R denotes Precision and Recall, respectively.

## 4.2 AGE ESTIMATE TASK

### 4.2.1 AFAD AND MORPH

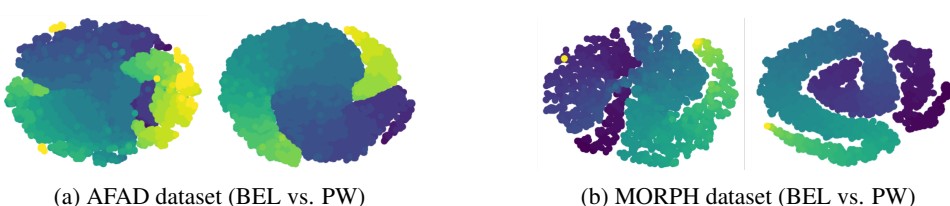

(a) AFAD dataset (BEL vs. PW)    (b) MORPH dataset (BEL vs. PW)

Figure 4: Visualization of BEL (left) vs. PW (right) models on (a) AFAD and (b) MORPH datasets.

Deval Shah reported BEL with ResNet50 Shah et al. (2022); for a fair comparison with prior work Niu et al. (2016); Cao et al. (2020), we re-implemented it using a unified ResNet34 backbone. BEL is the strongest baseline, and replacing only the final layer with our *Staircase* head (additive/subtractive) yields consistent gains (Table 2). Switching $L_{\mathrm{MAE}}$ to the mutual-information loss $L_{\mathrm{MI}}$ further improves results. On **AFAD**, the PW-MI variant achieves better scores: MAE 2.948 ($-5.8\%$ vs.BEL 3.13), CS5 0.856 ($+7.0\%$ vs.BEL 0.80). Similarly, On **MORPH**, the PW-MI variant achieves better scores: MAE 2.217 ($-4.8\%$ vs.BEL 2.33), CS5 0.925 ($+1.6\%$ vs.BEL 0.91). Latent-space visualizations (Figs. 4a–4b) show that PW-MI aligns more closely with the ground truth. Visualize the qualitative cases provided in Appendix R.

Table 2: Evaluate the proposed method against the best baseline on AFAD and MORPH2 dataset.

| | AFAD | | MORPH | |
|---|---|---|---|---|
| Model | MAE↓ | CS5↑ | MAE↓ | CS5↑ |
| BEL | 3.13 | 0.80 | 2.33 | 0.91 |
| PW-MI | $-5.8\%$ | $+7.0\%$ | $-4.8\%$ | $+1.6\%$ |

The table shows the improvement percentages relative to BEL Shah et al. (2022).

### 4.2.2 ABALONE

Unimodal Cardoso et al. (2025) enforces unimodality via UnimodalNet and a Wasserstein regularizer. On *Abalone* (their protocol; WU-KLDIV/WU-Wass; output replaced by our Staircase), all our variants w/ and w/o $L_{MI}$ outperform WU-Wass on all metrics (Table 3). AS-SS-MI achieved the best results. Acc 60.93 (+5.3%), MAE 0.475 (-10.4%), QWK (+9.4%), $\tau$ (+6.1%) and ZME (+58.2%).

Table 3: Evaluate the proposed method against the best baseline on ABALONE dataset.

| | ABALONE | | | | |
|---|---|---|---|---|---|
| Model | Acc% ↑ | MAE ↓ | QWK ↑ | $\tau$% ↑ | ZME → 0 |
| | | | Baseline | | |
| WU-KLDIV | $57.9_{2.6}$ | $0.54_{0.02}$ | $62.4_{2.1}$ | $63.1_{2.5}$ | $-0.18_{0.02}$ |
| WU-Wass | $57.9_{2.4}$ | $0.53_{0.02}$ | $63.3_{1.6}$ | $63.2_{2.1}$ | $-0.17_{0.02}$ |
| AS-SS-MI | +5.3% | -10.4% | +9.4% | +6.1% | +58.2% |

The table shows the improvement percentages relative to WU-Wass Cardoso et al. (2025).

## 4.3 MALIGNANT TUMOR DETECTION TASK

Using the same backbone of MetaOrdinal Lei et al. (2022), we replace the final layer with our Staircase activation. On BUSI (Table 4), Staircase consistently surpasses MetaOrdinal, indicating cross-domain generalization. The piecewise variant performs best: F1 = 0.907 with $L_{\mathrm{MAE}}$ (+17.0%) and 0.925 with $L_{\mathrm{MI}}$ (+18.6%). With $L_{\mathrm{MI}}$, per-class F1 also improves, especially for Normal and Benign.

Table 4: Evaluate the proposed method against the baseline on BUSI dataset.

| | BUSI (Overall) | | | | BUSI (Per-class F1) | | |
|---|---|---|---|---|---|---|---|
| Model | Precision↑ | Recall↑ | F1-Score↑ | Acc↑ | Normal↑ | Benign↑ | Malignant↑ |
| CORE | − | − | − | 0.82 | − | − | − |
| MORF | 0.82 | 0.77 | 0.78 | 0.80 | 0.682 | 0.845 | − |
| PW-MI | +14.7% | +19.4% | +19.4% | +14.5% | 0.959 | 0.934 | 0.881 |

− for CORE Lei et al. (2024) indicates that the corresponding result is not available, as the original paper did not provide it. The table shows the improvement percentages relative to MORF Lei et al. (2022).

## 4.4 MONOCULAR DEPTH TASK

**Protocol.** We adopt the scale-invariant loss function $L = \mathrm{SilogLoss} + L_{MI}$ on both datasets. For KITTI, we replace GLPDepth's final range scaling (multiplying a normalized map by a fixed max depth, e.g., 80 m) with a direct Staircase activation on the network output, removing explicit range assumptions. For NYUv2, we replace the OE regularizer Zhang et al. (2023) with Staircase activations under the same training protocol (SilogLoss Eigen et al. (2014) with optional $L_{MI}$).

**Results on KITTI.** With identical backbones, all Staircase variants improve both threshold accuracies ($\delta_1, \delta_2, \delta_3$) and error metrics (AbsRel, RMSE, RMSLE); see Table 5. The *AS-HS-MI* variant yields the strongest overall gains: $\delta_1$ +0.20%, AbsRel −3.50%, RMSE −2.10%, and RMSLE −2.30%, highlighting the benefit of the additive scheme with HardSigmoid and the complementary role of $L_{MI}$ to SilogLoss in achieving scale-robust depth.

**Results on NYUv2.** Staircase consistently outperforms OE across $\delta_1/\delta_2/\delta_3$ and AbsRel/RMSE/RMSLE; see Table 5. *NO-SS-MI* reaches up to +3.7% in $\delta_1$ and −6.6% in AbsRel. Fig. 5 presented qualitative examples. Qualitative examples are provided in Appendix R.

Table 5: Evaluate the proposed method against the baseline on KITTI and NYUv2 dataset.

| | KITTI | | | | | |
|---|---|---|---|---|---|---|
| Model | $\delta_1$↑ | $\delta_2$↑ | $\delta_3$↑ | AbsRel↓ | RMSE↓ | RMSLE↓ |
| GLPDepth | 0.967 | 0.996 | 0.999 | 0.057 | 2.297 | 0.086 |
| AS-HS-MI | +0.20% | +0.00% | +0.00% | −3.50% | −2.10% | −2.30% |
| | NYUv2 | | | | | |
| OE | 0.537 | 0.832 | 0.948 | 0.271 | 0.849 | 0.313 |
| NO-SS-MI | +3.7% | +1.4% | +0.5% | −6.6% | −2.9% | −3.5% |

The table shows the improvement percentages relative to GLPDepth Kim et al. (2022) and OrdinalEntropy (OE) Zhang et al. (2023).

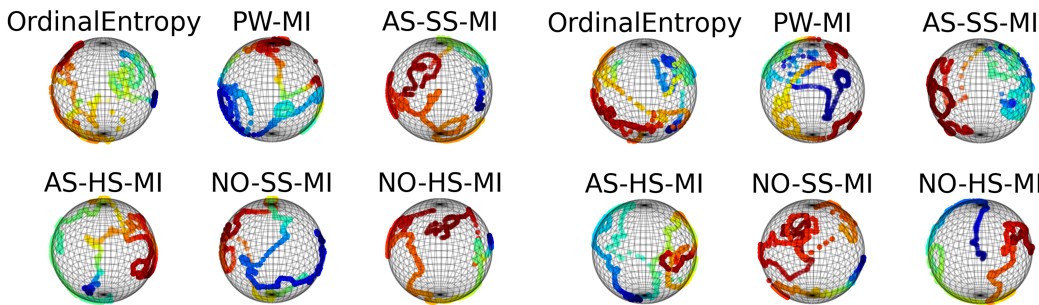

Figure 5: Latent embeddings of a single NYUv2 test image are projected onto the unit sphere using t-SNE, visualized for six different methods.

## 4.5 SENTIMENT ANALYSIS TASK

**Sentence-level sentiment (NLP).** Beyond raw data and images, we evaluate sentence-level sentiment to test cross-modal applicability. Following OLL Castagnos et al. (2022), we adopt BERT-Tiny as the backbone and replace its final layer with our Staircase activations. Both additive and subtraction variants consistently surpass OLL on standard metrics (Acc., Macro-F1), indicating that Staircase heads better capture ordinal sentiment intensity than conventional classifiers. The histogram in Appendix R Fig.11 shows that OLL1.5 Castagnos et al. (2022) makes virtually no predictions for *class 0*; this likely relates to BERT-Tiny calibration and is outside our scope. Under the same backbone, Staircase heads restore coverage across all labels and yield clear gains for *class 0*, *class 1*, *class 3*, and *class 4*.

Table 6: Evaluate the proposed method against the baseline on SST-5 dataset.

| | SST-5 | | |
| --- | --- | --- | --- |
| Model | MAE $\downarrow$ | MSE $\downarrow$ | $\tau_b$ $\uparrow$ |
| OL | 0.739 | 1.081 | 0.544 |
| PW-MI | $-4.2\%$ | $-3.8\%$ | $+4.0\%$ |

The table shows the improvement percentages relative to OLL Castagnos et al. (2022).

## 5 CONCLUSION AND FUTURE WORK

**Conclusion and Outlook.** We tackle the degeneration–saturation dilemma with multi-level, parameterized *Staircase* activations. Three data-driven remedies are introduced: (i) noise injection, (ii) an ascending monotone term, and (iii) an increasing piecewise-linear (PW) form. Gradient and mutual-information analyses motivate the learning rule and the regularizer $L_{MI}$.

**Effectiveness.** Across diverse datasets and real-world settings, Staircase heads deliver consistent improvements in ordinal regression, regardless of modality, output format, number of categories, or data scale—demonstrating competitive and versatile performance.

**Time Complexity.** We provided the inference efficiency of various activation functions in Appendix S.

**Limitations and Future Work.** Under distribution shift Rebbapragada et al. (2015); Quinonero-Candela et al. (2008), learned Staircase thresholds require adaptation. We will explore reinforcement-learning controllers to update thresholds online without domain-specific heuristics or full network retraining. Moreover, a systematic study of convergence and optimization behavior across different backbones remains open and is left for future work.

**LLM for writing assistance.** We used a large language model solely for light copyediting (grammar, wording, and clarity); all ideas, analyses, and results are the authors' own.

**Code Availability.** We will release the code (github) upon acceptance.

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
