# A    APPENDIX OF STEP ACTIVATION AND DERIVATIVE

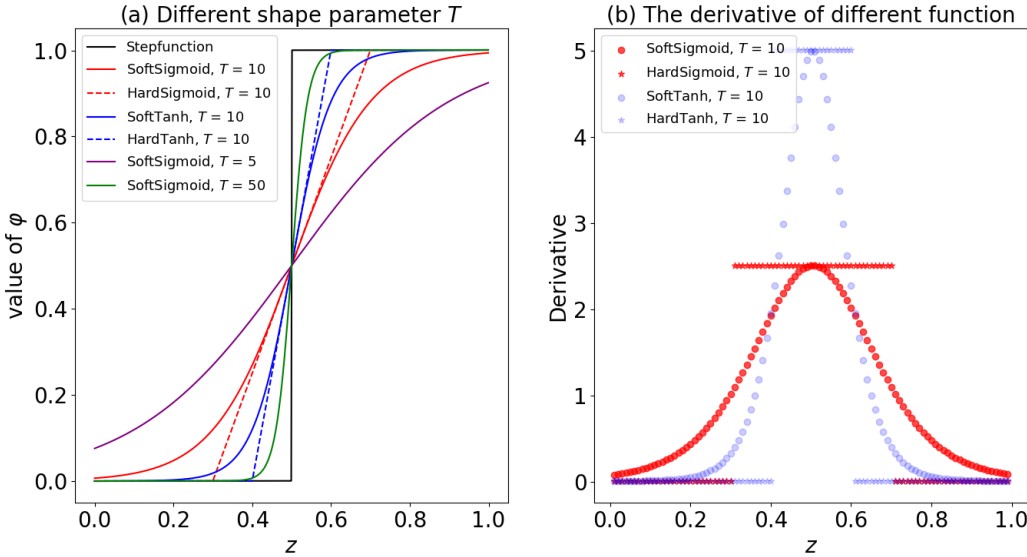

Figure 1: (a) Step activation function and various approximations. (b) Derivatives of step activation function approximations.

# B    SOME UNZIPPED VERSION OF DEFINITIONS

In contrast to additive strategies that perturb activations locally, a more fundamental solution is to reshape the activation function itself to eliminate saturation altogether. We propose a piecewise linear reformulation of the staircase activation, where saturated flat regions are removed and replaced with continuous linear segments between threshold levels. This subtraction-based design guarantees that the activation function maintains a strictly nonzero gradient across the entire operating range, effectively preventing gradient vanishing and degeneration.

$$
\psi_{\Xi,T}^{PW}(z) = \begin{cases}
-0.5, & \text{if } z \in (-\infty, 0] \\
\frac{\left(z - \frac{\xi_1 + 0}{2}\right)}{\xi_1 - 0} + 0, & \text{if } z \in (0, \xi_1] \\
\vdots & \vdots \\
\frac{\left(z - \frac{\xi_{k+1} + \xi_k}{2}\right)}{\xi_{k+1} - \xi_k} + k, & \text{if } z \in (\xi_k, \xi_{k+1}], k \in \{1, ..., K\} \\
\vdots & \vdots \\
\frac{\left(z - \frac{1 + \xi_{K-1}}{2}\right)}{1 - \xi_{K-1}} + K - 1, & \text{if } z \in (\xi_{K-1}, 1] \\
K - 0.5, & \text{if } z \in (1, \infty)
\end{cases} \tag{1}
$$

# C    APPENDIX OF MORE RELATED WORKS

## C.1    BACKGROUND ON ORDINAL REGRESSION

Ordinal regression—also called ordinal classification—dates back to statistical methods in the 1980s McCullagh (1980). It appears frequently in the social sciences (e.g., modeling levels of human preference, class rank) and in psychology Bürkner & Vuorre (2019); Bhargava (2005). Because ordinal variables are discrete labels with an inherent order but without defined inter-class distances, standard classification and regression models are not directly suitable Satake et al. (2018). Consequently, numerous approaches have been proposed, including SVM-based methods Wu et al. (2003); Chu & Keerthi (2007), Gaussian-process methods Chu et al. (2005); Liu et al. (2019), and

label- or threshold-based methods Cao et al. (2020); Lin & Li (2006). Label encodings such as NNRank Cheng et al. (2008) and BEL Shah et al. (2022) are attractive due to their architectural simplicity.

## C.2  RECENT OTHER DEEP LEARNING APPROACHES FOR ORDINAL CLASSIFICATION

Beyond activation function design, the deep learning community has explored various architectural and training strategies specifically for ordinal tasks. Recent advances include specialized loss functions that explicitly model ordinal relationships, such as ranking-based losses that minimize violations of ordinal constraints Gutiérrez et al. (2015) and earth mover's distance losses that penalize predictions proportionally to their ordinal distance from ground truth Hou et al. (2016). Distribution-based approaches enforce unimodal probability distributions over ordinal classes, ensuring that predictions respect the natural ordering Beckham & Pal (2017). Soft labeling techniques assign continuous targets that reflect ordinal proximity, allowing models to learn from the inherent structure of ordered categories Vargas et al. (2023). Recent work also explores curriculum learning approaches that present training examples in order of increasing ordinal difficulty Platanios et al. (2019). ORDAC proposes a data-centric method for ordinal image classification with noisy labels that uses label-distribution learning and cross-fold predictions to adaptively correct each sample's mean and standard deviation—rather than discarding data—yielding strong results on age estimation and diabetic retinopathy benchmarks. Sedighi Moghaddam & Mohammadi (2025). dlordinal proposed open-source, PyTorch-based Python package that unifies state-of-the-art deep ordinal classification methods—losses, output layers, soft-labeling schemes, and evaluation metrics—into a single, reproducible framework. Bérchez-Moreno et al. (2025). SLACE introduces a provably monotone and balance-sensitive ordinal-learning loss (Soft-Labels Accumulating Cross-Entropy) and shows state-of-the-art results on tabular ordinal regression benchmarks Nachmani et al. (2025).

## C.3  SATURATION BEHAVIOR: EVIDENCE AND PRIOR REMEDIES

Staircase functions capture multiclass, order-preserving behavior and predate deep learning. Several works explored staircase-like activations within neural networks Wang et al. (2019); Tóth & Gosztolya (2004) but encountered the *saturation behavior problem* Gulcehre et al. (2016); Chen et al. (2017), where gradients vanish or become undefined in saturated regions. Prior studies report that saturation harms both information capacity and trainability Rakitianskaia & Engelbrecht (2015); e.g., `tanh` suffers vanishing gradients Wang et al. (2019), and sigmoid-type functions exhibit sharp gradient damping, slow convergence, and non-zero-centered outputs Nwankpa et al. (2021); similar observations are made by Xu Xu et al. (2016). Tóth and Gosztolya Tóth & Gosztolya (2004) showed networks using a literal staircase failed to converge under backpropagation, whereas ramp-like or sigmoid activations trained successfully. A complementary strategy is to shrink saturation regions by steepening slopes Hu et al. (2019); Boskovitz & Guterman (2002), since steeper sigmoids reduce near-constant regimes that block gradient flow Boskovitz & Guterman (2002). Appendix Fig. **??** illustrates typical saturation at extremes for sigmoid/tanh.

## C.4  NOISY ACTIVATIONS AND ALTERNATIVE TRAINING SIGNALS

A line of work, rooted in Guozhong's classic results An (1996), shows that certain forms of noise can improve generalization in classification. Motivated here by trainability rather than regularization alone, we consider *noisy staircase activations*—with learnable thresholds and slopes—to mitigate saturation in sigmoid/softmax regimes Gulcehre et al. (2016); Chen et al. (2017), akin to Gaussian error units with learnable variance Duan et al. (2024). Guozhong An (1996) systematically examined adding noise to inputs, outputs, weights, and updates, concluding that weight noise aids classification generalization, whereas other placements often yield limited benefit. Neelakantan et al. Neelakantan et al. (2015), inspired by Welling & Teh (2011), propose time-dependent Gaussian noise per step, $g_t \leftarrow g_t + \mathcal{N}(0, \sigma_t^2)$, $\sigma_t^2 = \frac{\eta}{(1+t)^\gamma}$, a low-overhead technique that can improve optimization. As an orthogonal alternative to backprop through zero-derivative steps, *target propagation* Cun (1986); Friesen & Domingos (2018) propagates targets instead of gradients.

## C.5 REMARK ON BENCHMARKS

TOC-UCO Ayllón-Gavilán et al. (2025) compiles many tabular ordinal datasets, but its benchmarks rely largely on classical ML (XGBoost, Random Forest) and shallow MLPs rather than deep architectures where activation-function design is critical. Given its arXiv status, comparisons in the main text prioritize peer-reviewed baselines and deep architectures.

## D APPENDIX OF SIGMOID TRANSFORM TO TANH

We set ordinal regression indices starting from 0, and align the Tanh-based series Staircase activation function with the Sigmoid-Based series by applying an adjustment of *adding 1 and dividing by 2* to shift the Tanh-based functions. Yet, upon incorporating the trainable parameter $T$ into our considerations, we realize that the Tanh-based Staircase functions becomes equivalent to the Sigmoid-Based functions when $T$ is reduced by half. That is :

$$\psi^{ss,hs}(z,\Xi,T) \equiv \psi^{st,ht}(z,\Xi,\frac{T}{2}) \tag{2}$$

## E MORE CHALLENGES WITH PARAMETERIZED STAIRCASES

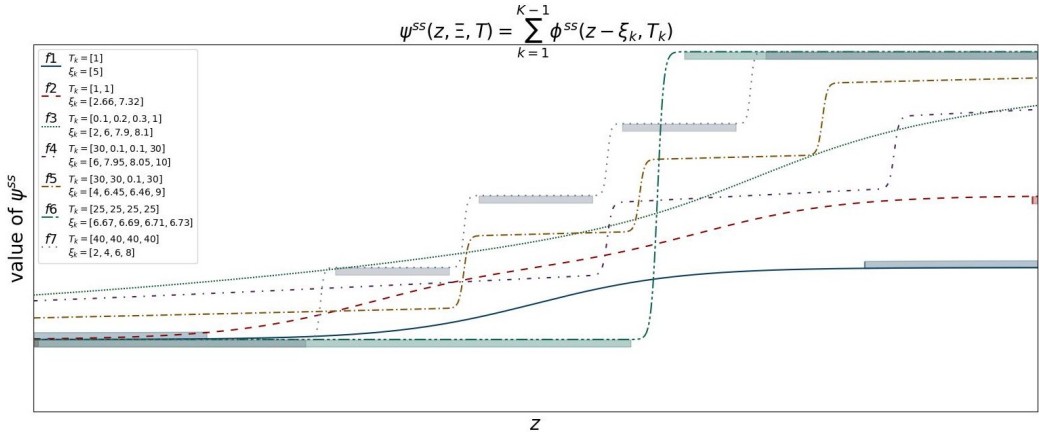

Figure 2: Shaded bands beneath each curve denote saturated domains where the derivative (gradient) is below $10^{-2}$. Here, $f_1$ denotes a center-shifted standard sigmoid; $f_2$ denote 3-stage and $f_3$ to $f_6$ denotes 5-stage saturated staircases, respectively; and $f_7$ denotes a non-saturated 5-stage staircase.

## F APPENDIX OF PROOF OF LEMMA 1 (NEAR-SATURATION LOWER BOUND)

*Proof.* (Sketch.) To convey the main idea, consider a neural network trained using the mean absolute error (MAE) loss function with the output layer employing $\psi^{ss}$. Using the chain rule, the backpropagation of the layer ($L$) can be expressed as follows:

$$\begin{aligned}
z^L &= w_1^L z^{L-1} + w_0^L \\
\hat{y} &= \psi^{ss}(z^L, \Xi, T) \\
\mathcal{L}oss &= |\hat{y} - y|
\end{aligned} \tag{3}$$

where $z^l$ denotes the output of the layer $l \in [L]$, and $w_i^L \in \mathbb{R}$ for $i \in \{0,1\}$ are the weights. The subscript $i$ may be omitted for convenience. For a Soft-Sigmoid Staircase activation function:

$$\frac{\partial \mathcal{L}oss}{\partial w^L} = \frac{\partial z^L}{\partial w^L} \frac{\partial \hat{y}}{\partial z^L} \frac{\partial \mathcal{L}oss}{\partial \hat{y}}$$

$$= z^{L-1} \times \sum_{k=1}^{K-1} T_k \times \phi^{ss}(z^L - \xi_k)(1 - \phi^{ss}(z^L - \xi_k)) \quad (4)$$

$$\times \frac{\hat{y} - y}{|\hat{y} - y|}.$$

Let the superscript $(t+1)$ denote the corresponding result of $(t+1)$-th iteration, and update the weights:

$$w^{L,(t+1)} = w^{L,(t)} - \gamma \times \frac{\partial \mathcal{L}oss}{\partial w^L} \quad (5)$$

The idea is simple: negligible value of $|\frac{d\psi^{ss}}{dz^L}|$, i.e., $\leq \epsilon^2$, implies $|\frac{\partial \mathcal{L}oss}{\partial w^L}|$, $|w^{L,(t+1)} - w^{L,(t)}|$, $|z^{L,(t+1)} - z^{L,(t)}|$, and $|\hat{y}^{(t+1)} - \hat{y}^{(t)}|$ are all extremely small. In particular,

$$|\hat{y}^{(t+1)} - \hat{y}^{(t)}| = |\psi^{ss}(z^{L,(t+1)}) - \psi^{ss}(z^{L,(t)})| < \gamma\epsilon. \quad (6)$$

Hence if $S$ iterations can fix the prediction error, then

$$1 \leq |y - \hat{y}^{(1)}| \leq \sum_{t=1}^{S} |\hat{y}^{(t+1)} - \hat{y}^{(t)}| < S\gamma\epsilon, \quad (7)$$

which means $S > (\gamma\epsilon)^{-1}$, i.e., $\Omega((\gamma\epsilon)^{-1})$. $\qquad\square$

Observe that when $z^L \in \mathcal{SD}_{\psi,0}$ it is clear that $\frac{\partial \hat{y}}{\partial z^L} = 0$ in Eq. (13) and then $\frac{\partial \mathcal{L}oss}{\partial w^L} = 0$, indicating that the trainable weights cannot be updated. This lemma emphasizes that not only does a gradient of zero have a negative impact, but a gradient that is excessively small also poses detrimental effects.

# G APPENDIX OF NOISY STAIRCASE ACTIVATION FUNCTION

The (binary) noisy activation function Gulcehre et al. (2016) can be rewritten in the following form:

$$\begin{aligned}
\triangle &= \phi^{hs}(z) - u^s(z) \\
\sigma_{\triangle,\beta,p} &= \beta(\phi^{ss}(p\triangle) - 0.5)^2 \\
\phi^{noisy}_{\triangle,\alpha,\beta,p}(z) &= \alpha \times \phi^{hs}(z) + (1 - \alpha) \times u^s(z) \\
&\quad + d(z) \times \sigma_{\triangle,\beta,p} \times \mathcal{O}
\end{aligned} \quad (8)$$

where $\phi^{hs}$, $u^s$, and $\phi^{ss}$ are defined in Sec.3.1 with $T = 1$. Additionally, the scalar hyperparameter $\beta$ is designed for controlling the scale of the standard deviation of the added noise, and $p$ is a learnable scalar parameter. The hyperparameter $\alpha$ is expected to be near 1 and is set for the function $d(z) = -sgn(z)sgn(1 - \alpha)$, which helps to push saturated units towards a non-saturated state. Finally, the noise component $\mathcal{O} \sim N(0, 1)$, drawn from the normal distribution with mean 0 and standard deviation 1. By the similar extension with parameters $\Xi, T$ shown in Eq.3,

We inject noise in *non-saturation behavior region*.

$$u_k^{ss,hs} = (0.25 \times T(z - \xi_k) + 0.5) \quad (9)$$

$$\triangle_k^{ss,hs} = (\phi^{ss,hs}(z - \xi_k, T) - u_k^{ss,hs}) \quad (10)$$

$$\psi^{noisy;ss,hs}(z, \alpha, \Xi, T, \beta) = \alpha \times \sum_{k=0}^{K-1} \phi^{ss,hs}(z, \xi_k, T)$$

$$+ (1 - \alpha) \times \sum_{k=0}^{K-1} u_k^{ss,hs} \qquad (11)$$

$$+ \prod_{k=0}^{K-1} \sigma(\triangle_k^{ss,hs}, \beta, p) \times \mathcal{O}$$

We notice that $\alpha$ has a rotation property which makes rotate both left and right ends, when $\alpha > 1$, it leads the Staircase activation functions to appear to be a property of downhill, and has a reverse property when $\alpha < 1$, In particular, we ignore the direction term $d(x)$, since we don't need to push the saturated unit towards a non-saturated state, it will lead to prediction always lower.

## H  APPENDIX OF ANALYSIS WITH GRADIENT

The three proposed strategies—Noisy Staircase Activation, Ascending Staircase Activation, and Piecewise Linear Staircase Activation—are all designed to maintain effective gradient flow across the activation domain, thereby preventing saturation-induced training difficulties.

In this appendix, we theoretically analyze two key aspects: (1) whether these strategies guarantee non-vanishing gradients within the staircase intervals, and (2) whether the learnable thresholds can be effectively optimized during training.

To formally characterize the gradient behavior, we establish the following lemma:

**Lemma 2.** *Fro given $\Xi, T$, there exists $\epsilon^* > 0$ such that the proposed methods can prevent the saturation phenomenon for $z \in (\xi_1, \xi_K)$.*

*Proof.* (Sketch.)  First note that the gradient of $\psi_{\Xi,T}^{AS;ss,hs}(z)$ and $\psi_{\Xi,T}^{PW}(z)$ is positive since $\frac{d}{dz^L} sin^{-1}(\frac{z^L}{\mu}) = \frac{1}{\sqrt{\mu^2 - (z^L)^2}} > 0$ and $\frac{d}{dz^L} \psi_{\Xi,T}^{PW} = 1/(\xi_{k+1} - \xi_k) > 0$ for $k \in [K-1]$. Furthermore, *in a single training epoch*, the gradient of $\psi_{\Xi,T}^{NO;ss,hs}(z)$ is non-zero almost everywhere since $\mathcal{O} \sim N(0, 1)$. Set $\epsilon^*$ as the minimum value of the gradient of $\psi \in \{\psi_{\Xi,T}^{AS;ss,hs}, \psi_{\Xi,T}^{PW}\}$, then it is straightforward that the interval-length of $\mathcal{SD}_{\psi,\epsilon} \cap (\xi_1, \xi_K)$ is zero. $\qquad\square$

This lemma implies that the three methods can prevent the conditions of Lemma 1. The following lemma proves that thresholds can be learned from training data.

**Lemma 3.** *Let $0 < \xi_1 < \cdots < \xi_{K-1} < 1$ be the unknown inherent thresholds of the latent space data $\{z\}$. Then the minimum of mean absolute error (or mean squared error) attains if the estimated thresholds equal $\xi_k$ for all $k \in [K]$.*

*Proof.* (Sketch.) Since each point $z$ is possible to contribute the absolute error $(AE)$ $|y(z) - \hat{y}(z)|$ or the squared error $(SE)$ $(y(z) - \hat{y}(z))^2$, the total error can be estimated as $\int_{z \in [0,1]} |y(z) - \hat{y}(z)| dz$ or $\int_{z \in [0,1]} (y(z) - \hat{y}(z))^2 dz$. For clearness and simplicity, consider the case of $K = 3$. That is to say, $0 < \xi_1 < \xi_2 < 1$ are the unknown inherent thresholds for the ordinal classification corresponding to $\{y_k\}$ on the interval $[0, 1]$ of the $z$-axis. Let $0 < \eta_1 < \eta_2 < 1$ be the trainable thresholds estimating $\xi_i$ for the Piece-wise activation function which can be represented as $\hat{y} = \frac{\left(z - \frac{\eta_{k+1} + \eta_k}{2}\right)}{\eta_{k+1} - \eta_k} + 1$, for $z \in$

$(\eta_k, \eta_{k+1}],$, where $k = \{0, 1, 2\}$, $\eta_0 \triangleq 0$, and $\eta_3 \triangleq 1$. Hence,

$$AE(\eta_1, \eta_2) := \int_{z \in [0,1]} |y(z) - \hat{y}(z)| dz$$

$$= \begin{cases} \frac{1}{4} + \frac{(\eta_1 - \xi_1)^2}{\eta_1} + \frac{(\eta_2 - \xi_2)^2}{\eta_2 - \eta_1} & \text{if } \eta_1 > \xi_1, \eta_2 > \xi_2; \\ \frac{1}{4} + \frac{(\eta_1 - \xi_1)^2}{\eta_1} + \frac{(\eta_2 - \xi_2)^2}{1 - \eta_2}, & \text{if } \eta_1 > \xi_1, \eta_2 < \xi_2; \\ \frac{1}{4} + \frac{(\eta_1 - \xi_1)^2}{\eta_2 - \eta_1} + \frac{(\eta_2 - \xi_2)^2}{1 - \eta_2}, & \text{if } \eta_1 < \xi_1, \eta_2 < \xi_2; \\ \frac{1}{4} + \frac{(\eta_1 - \xi_1)^2}{\eta_2 - \eta_1} + \frac{(\eta_2 - \xi_2)^2}{\eta_2 - \eta_1}, & \text{if } \eta_1 > \xi_1, \eta_2 > \xi_2. \end{cases}$$

Obviously, for each case, both $AE(\eta_1, \eta_2)$ (or $SE(\eta_1, \eta_2)$), the global minimum is achieved as $\eta_1 = \xi_1, \eta_2 = \xi_2$. A similar argument, along with multi-variable calculus, can be applied to reach the same conclusion in the general case for a larger number of classes (K¿3).. $\square$

This Lemma implies the learning method. For a given training dataset $\{(z_n, y_n)\}_{n \in [N]}$ in latent space, in order to learn $\{\xi_k\}_{k \in [K]}$ by the gradient descent, $\mathcal{L}_\xi := \left( \frac{1}{N} \sum_{n=1}^{N} |y(z_n) - \hat{y}(z_n)| \right) - \frac{1}{4}$ can be a part of the total training loss since $N$ large enough $\frac{1}{N} \sum_{n=1}^{N} |y(z_n) - \hat{y}(z_n)| \approx \int_{z \in [0,1]} |y(z_n) - \hat{y}(z_n)| dz$.

# I APPENDIX OF DETAILED PROOFS OF LEMMAS

**Lemma 4.** *Consider the Staircase activation function $\psi$ serving as the output layer of a given deep learning model $F_\Theta$ and having a non-zero length $\frac{1}{\epsilon^2}$-near saturation domain. Let $\gamma$ denote the learning rate. For a training sample $(x, y)$ that $\hat{y} = F_\Theta(x) = \psi^{ss}(z)$, and $|\hat{y} - y| \geq 1$, then the training process based on the gradient descent needs (at least) $\Omega((\gamma\epsilon)^{-1})$ steps of updating the trainable weights $\Theta$ s.t. $\hat{y} = y$, if $z(x)$ fall in the $\frac{1}{\epsilon^2}$-near saturation domain $\mathcal{SD}_{\psi,\epsilon^2}$.*

*Proof.* To convey the main idea, consider a neural network trained using the mean absolute error (MAE) loss function with the output layer employing $\psi^{ss}$. Using the chain rule, the backpropagation of the layer ($L$) can be expressed as follows:

$$\begin{aligned} z^L &= w_1^L z^{L-1} + w_0^L \\ \hat{y} &= \psi^{ss}(z^L, \Xi, T) \\ \mathcal{L}oss &= |\hat{y} - y| \end{aligned} \tag{12}$$

where $z^l$ denotes the output of the layer $l \in [L]$, and $w_i^L \in \mathbb{R}$ for $i \in \{0, 1\}$ are the weights. The subscript $i$ may be omitted for convenience. For a Soft-Sigmoid Staircase activation function:

$$\begin{aligned} \frac{\partial \mathcal{L}oss}{\partial w^L} &= \frac{\partial z^L}{\partial w^L} \frac{\partial \hat{y}}{\partial z^L} \frac{\partial \mathcal{L}oss}{\partial \hat{y}} \\ &= z^{L-1} \times \sum_{k=1}^{K-1} T_k \times \phi^{ss}(z^L - \xi_k)(1 - \phi^{ss}(z^L - \xi_k)) \\ &\quad \times \frac{\hat{y} - y}{|\hat{y} - y|}. \end{aligned} \tag{13}$$

Let the superscript $(t + 1)$ denote the corresponding result of $(t + 1)$-th iteration, and update the weights:

$$w^{L,(t+1)} = w^{L,(t)} - \gamma \times \frac{\partial \mathcal{L}oss}{\partial w^L} \tag{14}$$

The idea is simple: negligible value of $|\frac{d\psi^{ss}}{dz^L}|$, i.e., $\leq \epsilon^2$, implies $|\frac{\partial \mathcal{L}oss}{\partial w^L}|$, $|w^{L,(t+1)} - w^{L,(t)}|$, $|z^{L,(t+1)} - z^{L,(t)}|$, and $|\hat{y}^{(t+1)} - \hat{y}^{(t)}|$ are all extremely small. In particular, obviously $\mathcal{SD}_{\psi,\epsilon^2} \subset$

$\mathcal{SD}_{\psi,\epsilon}$, then

$$
\begin{aligned}
|\hat{y}^{(t+1)} - \hat{y}^{(t)}| &= |\phi^{ss}(z^{L,(t+1)}) - \phi^{ss}(z^{L,(t)})| \\
&\leq \left| \frac{d\phi^{ss}}{dz}(z^*) \times \left( z^{L,(t+1)} - z^{L,(t)} \right) \right| \\
&\leq \left| \frac{d\phi^{ss}}{dz}(z^*) \times \gamma \times \frac{\partial \mathcal{L}oss}{\partial w^L} \right| \\
&< \epsilon \times \gamma \times \frac{K \times T^*}{e^{T^* \times (z^{L,(t)} - \xi_k)}} \\
&< \gamma\epsilon,
\end{aligned}
\tag{15}
$$

where $z^* = \arg\max\{\frac{d\phi^{ss}}{dz}(z) : z \in Interval(z^{L,(t+1)}, z^{L,(t)})\}$, $Interval(z^{L,(t+1)}, z^{L,(t)})$ is the interval defined by the boundary $z^{L,(t+1)}$ and $z^{L,(t)}$, and $T^* = \max\{T_k : k \in [K]\}$. Hence if $S$ iterations can fix the prediction error, i.e.,

$$
\hat{y}^{(S)} = y,
$$

then

$$
\begin{aligned}
1 &\leq |y - \hat{y}^{(1)}| \\
&= |\hat{y}^{(S)} - \hat{y}^{(1)}| \\
&\leq |\hat{y}^{(S)} - \hat{y}^{(S-1)}| + \ldots\ldots + |\hat{y}^{(2)} - \hat{y}^{(1)}| \\
&= \sum_{t=1}^{S} |\hat{y}^{(t+1)} - \hat{y}^{(t)}| \\
&< S\gamma\epsilon,
\end{aligned}
\tag{16}
$$

which means $S > (\gamma\epsilon)^{-1}$, i.e., $\Omega((\gamma\epsilon)^{-1})$. $\qquad\square$

**Lemma 5.** *Let $0 < \xi_1 < \cdots < \xi_{K-1} < 1$ be the unknown inherent thresholds of the latent space data $\{z\}$. Then the minimum of mean absolute error (or mean squared error) attends if the estimated thresholds equal to $\xi_k$ for all $k \in [K]$.*

*Proof.* Without loss of generality, we can only consider $\{z^L\}$ on the interval $[0, 1]$ of the $z$-axis. Let $0 < \xi_1 < \cdots < \xi_N < 1$ be the sequence of unknown inherent thresholds for the ordinal classification corresponding to $\{y^L\}$ on the interval $[0, 1]$ of the $z$-axis. Let $0 < \eta_1 < \cdots < \eta_N < 1$ be the sequences of trainable thresholds for the Piece-wise activation function. That is, for each $i \in [N]$, $\eta_i$ is the estimation of $\xi_i$. Consider the step of finding $\eta_i$ that estimates $\xi_i$. Since $\{\xi_i\}_{i \in [N]}$ are unknown, the close form of PW activation function can only be defined by $\{\eta_i\}_{i \in [N]}$. In particular, we have $\hat{y} = \frac{1}{\eta_{i+1} - \eta_i}\left( z - \frac{\eta_{i+1} + \eta_i}{2} \right) + (i + 1)$, for $z \in (\eta_i, \eta_{i+1}]$.

$$
\hat{y} = \frac{\left( z - \frac{\eta_i + \eta_{i-1}}{2} \right)}{\eta_i - \eta_{i-1}} + (i - 1), \text{for } z \in (\eta_{i-1}, \eta_i],
$$

$$
\hat{y} = \frac{\left( z - \frac{\eta_{i+1} + \eta_i}{2} \right)}{\eta_{i+1} - \eta_i} + (i + 1), \text{for } z \in (\eta_i, \eta_{i+1}],
$$

$$
\hat{y} = \frac{\left( z - \frac{\eta_{i+2} + \eta_{i+1}}{2} \right)}{\eta_{i+2} - \eta_{i+1}} + (i + 2), \text{for } z \in (\eta_{i+1}, \eta_{i+2}]
$$

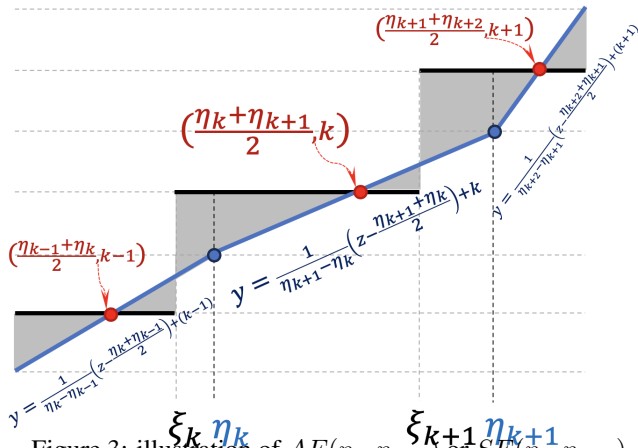

Figure 3: illustration of $AE(\eta_k, \eta_{k+1})$ or $SE(\eta_k, \eta_{k+1})$

The error in the training phases comes from the gray polygons shown in Fig. **??**. Since each point $z$ is possible to contribute the absolute error $|y(z) - \hat{y}(z)|$ or the squared error $(y(z) - \hat{y}(z))^2$, the total error can be estimated as $\int_{z \in [0,1]} |y(z) - \hat{y}(z)| dz$ or $\int_{z \in [0,1]} (y(z) - \hat{y}(z))^2 dz$. For clearness and simplicity, consider the case of $N = 2$. That is to say, $0 < \xi_1 < \xi_2 < 1$ are the unknown inherent threshold for the ordinal classification corresponding to $\{y^L\}$ on the interval $[0, 1]$ of the $z$-axis. Let $0 < \eta_1 < \eta_2 < 1$ be the trainable thresholds for the Piece-wise activation function which can be represented as

$$\hat{y} = \frac{\left(z - \frac{\eta_i + 0}{2}\right)}{\eta_1 - 0} + 0, \text{ for } z \in (0, \eta_1],$$

$$\hat{y} = \frac{\left(z - \frac{\eta_2 + \eta_1}{2}\right)}{\eta_2 - \eta_1} + 1, \text{ for } z \in (\eta_1, \eta_2],$$

$$\hat{y} = \frac{\left(z - \frac{1 + \eta_{i+1}}{2}\right)}{1 - \eta_2} + 2, \text{ for } z \in (\eta_2, 1]$$

Hence, if $\eta_1 > \xi_1$ and $\eta_2 > \xi_2$ then

$$
\begin{aligned}
AE(\eta_1, \eta_2) &:= \int_{z \in [0,1]} |y(z) - \hat{y}(z)| dz \\
&= \left\{ \frac{1}{2} \frac{\eta_1}{2} \frac{1}{2} + \frac{1}{2} \frac{\eta_2 - \eta_1}{2} \frac{1}{2} + \frac{1}{2} \frac{1 - \eta_2}{2} \frac{1}{2} \right\} \times 2 \\
&\quad + (\eta_1 - \xi_1) - \left\{ \left( \frac{2\xi_1 - \eta_1}{2\eta_1} + \frac{1}{2} \right)(\eta_1 - \xi_1) \right\} \times 2 \\
&\quad + (\eta_2 - \xi_2) - \left\{ \left( \frac{2\xi_2 - \eta_2 - \eta_1}{2(\eta_2 - \eta_1)} + \frac{1}{2} \right)(\eta_2 - \xi_2) \right\} \times 2 \\
&= \cdots \cdots \\
&= \frac{1}{4} + \frac{(\eta_1 - \xi_1)^2}{\eta_1} + \frac{(\eta_2 - \xi_2)^2}{\eta_2 - \eta_1}
\end{aligned}
$$

Also,

$$
AE(\eta_1, \eta_2)
= \begin{cases}
\frac{1}{4} + \frac{(\eta_1 - \xi_1)^2}{\eta_1} + \frac{(\eta_2 - \xi_2)^2}{1 - \eta_2}, & \text{if } \eta_1 > \xi_1, \eta_1 < \xi_1; \\
\frac{1}{4} + \frac{(\eta_1 - \xi_1)^2}{\eta_2 - \eta_1} + \frac{(\eta_2 - \xi_2)^2}{1 - \eta_2}, & \text{if } \eta_1 < \xi_1, \eta_1 < \xi_1; \\
\frac{1}{4} + \frac{(\eta_1 - \xi_1)^2}{\eta_2 - \eta_1} + \frac{(\eta_2 - \xi_2)^2}{\eta_2 - \eta_1}, & \text{if } \eta_1 > \xi_1, \eta_1 > \xi_1.
\end{cases}
$$

Furthermore, if $\eta_1 > \xi_1$ and $\eta_2 > \xi_2$ then we have

$$
\begin{aligned}
SE(\eta_1, \eta_2) &:= \int_{z \in [0,1]} (y(z) - \hat{y}(z))^2 dz \\
&= \int_{z \in [0,\eta_1]} \left( \frac{2z - \eta_1}{2\eta_1} \right)^2 dz + \int_{z \in [\eta_1, \eta_2]} \left( \frac{2z - \eta_2 - \eta_1}{2(\eta_2 - \eta_1)} \right)^2 dz \\
&\quad + \int_{z \in [\eta_2, 1]} \left( \frac{2z - 1 - \eta_1}{2(1 - \eta_2)} \right)^2 dz \\
&\quad + \int_{z \in [\xi_1, \eta_1]} \left( \frac{2z - \eta_1}{2\eta_1} - 1 \right)^2 - \left( \frac{2z - \eta_1}{2\eta_1} \right)^2 dz \\
&\quad + \int_{z \in [\xi_2, \eta_1]} \left( \frac{2z - \eta_2 - \eta_1}{2(\eta_2 - \eta_1)} - 1 \right)^2 - \left( \frac{2z - \eta_2 - \eta_1}{2(\eta_2 - \eta_1)} \right)^2 dz \\
&= \cdots\cdots \\
&= \frac{1}{12} + \frac{(\eta_1 - \xi_1)^2}{\eta_1} + \frac{(\eta_2 - \xi_2)^2}{\eta_2 - \eta_1}
\end{aligned}
$$

Also,

$$
SE(\eta_1, \eta_2)
= \begin{cases}
\frac{1}{12} + \frac{(\eta_1 - \xi_1)^2}{\eta_1} + \frac{(\eta_2 - \xi_2)^2}{1 - \eta_2}, & \text{if } \eta_1 > \xi_1, \eta_1 < \xi_1; \\
\frac{1}{12} + \frac{(\eta_1 - \xi_1)^2}{\eta_2 - \eta_1} + \frac{(\eta_2 - \xi_2)^2}{1 - \eta_2}, & \text{if } \eta_1 < \xi_1, \eta_1 < \xi_1; \\
\frac{1}{12} + \frac{(\eta_1 - \xi_1)^2}{\eta_2 - \eta_1} + \frac{(\eta_2 - \xi_2)^2}{\eta_2 - \eta_1}, & \text{if } \eta_1 > \xi_1, \eta_1 > \xi_1.
\end{cases}
$$

Obviously, for each case and for both $AE(\eta_1, \eta_2)$ and $SE(\eta_1, \eta_2)$, the global minimum is attended as $\eta_1 = \xi_1, \eta_2 = \xi_2$. $\qquad\square$

**Remark.** (1) In practice, the better the training effect the more concentrated $\{z\}$ near the location of the corresponding thresholds. (2) For given training dataset $\{(z_n, y_n)\}_{n \in [N]}$ in latent space, in order to learning $\{\xi_k\}_{k \in [K]}$ by the gradient decent, $\mathcal{L}_\xi := \left( \frac{1}{N} \sum_{n=1}^{N} |y(z_n) - \hat{y}(z_n)| \right) - \frac{1}{4}$ can be a part of total training loss since as $N$ large enough $\frac{1}{N} \sum_{n=1}^{N} |y(z_n) - \hat{y}(z_n)| \approx \int_{z \in [0,1]} |y(z_n) - \hat{y}(z_n)| dz$.

## J    APPENDIX OF MUTUAL-INFORMATION PERSPECTIVE ON THE STAIRCASE OBJECTIVE

Beyond gradient flow considerations, it is equally critical that the learned feature representations preserve sufficient information about the ordinal labels throughout training.

To this end, we analyze the connection between the proposed absolute-error-based loss and mutual information (MI) between features and labels, aiming to theoretically justify the stability and effectiveness of our approach.

Boudiaf et al. Boudiaf et al. (2020) emphasized that the process of minimizing cross-entropy loss can lead to maximization of mutual information between the feature and the label, which is defined as $I(Z;Y) = H(Z) - H(Z|Y)$. This is achieved by maximizing the entropy of the feature $H(Z)$ and minimizing the conditional entropy $H(Z|Y)$ through pairwise cross-entropy minimization.

Shihao et al. Zhang et al. (2023) demonstrated the relationship between mean squared error (MSE) loss and conditional entropy $H(Z|Y)$. Motivated by these studies, we adopt a similar approach and aim to establish the connection between the mean absolute error (MAE) utilized in our experiments and $H(Z|Y)$. Furthermore, we provide evidence that the Staircase activation function inherently supports maximizing $H(Z)$ by encouraging a more dispersed feature distribution.

**Theorem 1.** *Assume $(Z^c|Y) \sim Laplace(z_{c_k}, I)$, where $Z^c$ is the distribution of $z_{c_k}$ (center of class). Therefore, minimizing $L_{mae}$ with the Staircase activation function can serve as a substitute to minimize $H(Z|Y)$ without increasing $H(Z)$.*

*Proof.* (Sketch.) We use $\equiv$ to denote equivalence, the detailed proof can be found in Appendix D, where $\frac{1}{N} \sum_n^N ||z_n - z_{c_k}||$ can be interpreted as a conditional cross-entropy between $Z$ and $Z^c$.

$$
\begin{aligned}
\min L_{MAE} &= \min \frac{1}{N} \sum_n^N |S(z_n) - y_n| \\
&\equiv \min \frac{1}{N} \sum_n^N ||z_n - z_{c_k}||
\end{aligned}
\tag{17}
$$

This interpretation satisfies

$$
\begin{aligned}
CE(Z; Z^c|Y) &= -\mathbb{E}_{z \sim Z|Y}[log\, p_{Z^c|Y}(z)] \\
&\stackrel{mc}{\approx} \frac{-1}{N} \sum_n^N log(e^{(||z_n - z_{c_k}||)}) + \text{const} \\
&\stackrel{c}{=} \sum_n^N ||z_n - z_{c_k}||
\end{aligned}
\tag{18}
$$

In the provided context, where $\stackrel{c}{=}$ represents equivalence up to a multiplicative and additive constant, and $\stackrel{mc}{\approx}$ signifies Monte Carlo sampling from the $Z|Y$ distribution to substitute the expectation with the sample mean, we can subsequently demonstrate that

$$
\begin{aligned}
\min L_{MAE} &\equiv \min \frac{1}{N} \sum_n^N ||z_n - z_{c_k}|| \\
&\stackrel{c}{=} \min CE(Z; Z^c|Y) \\
&= \min H(Z|Y) + D_{KL}(Z||Z^c|Y)
\end{aligned}
\tag{19}
$$

This finding indicates that minimizing $L_{mae}$ using the Staircase activation function can serve as a substitute to minimize $H(Z|Y)$ without increasing $H(Z)$ when training converge of $D_{KL}(Z||Z^c|Y)$. Although in feature engineering it is generally assumed that the feature $z$ is based on a normal distribution Yang et al. (2021) in most cases,several studies have been conducted, with a focus on the distribution $z$, which is based on the Laplace distribution Liang & Zhang (2023); Dong et al. (2021); Barello et al. (2018); de Oliveira et al. (2020). In various scenarios of feature distribution, we have employed the MAE and reached the same conclusion as the MSE Zhang et al. (2023). The conclusion is to minimize $H(Z|Y)$ without increasing $H(Z)$.    $\square$

One distance-based approach for estimating the entropy $H(Z)$ of a feature $z$ is the meanNN entropy estimator Faivishevsky & Goldberger (2008), which is defined as follows:

$$\hat{H}(Z) = \frac{D}{N(N-1)} \sum_{n \neq m} log||z_n - z_m||^2 + const \quad (20)$$

The symbol $D$ is utilized to denote the dimension, and it is employed to estimate the differential entropy of the feature $z$. However, which is commonly used in high-dimensional space Faivishevsky & Goldberger (2010).

To maintain the ordinal characteristic of the data, we propose a novel loss term on the one-dimensional feature space $Z$ of the staircase activation function, which encourages maximizing entropy $H(Z)$, the loss term is defined as follows, where $\boldsymbol{z_k}$ and $\boldsymbol{z_{i+1}}$ belong to $y_k$ and $y_{k+1}$, respectively.

$$L_{Div} = -\frac{1}{K-1} \sum_{k=1}^{K-1} |max(\boldsymbol{z_k}) - min(\boldsymbol{z_{k+1}})| \quad (21)$$

Here, the loss term encourages maximizing the distance between class pairs during the training process while simultaneously considering the trade-off factor $\lambda_{Div}$ to ensure the preservation of the ordinal property. The final loss function as follows.

$$L_{MI} = L_{MAE} + \lambda_{Div} L_{Div} \quad (22)$$

By introducing the trade-off factor $\lambda_{Div}$, minimizing the loss function $L_{MI}$ with the Staircase activation function enables us to simultaneously maximize the mutual information $I(Z;Y)$ and preserve the ordinal property. We provid a detailed proof in Appendix E.

Intuitively, maximizing $H(Z)$ encourages features from different classes to occupy diverse regions of the feature space, preventing collapse. Minimizing $H(Z|Y)$ ensures that features belonging to the same class are tightly clustered. Thus, mutual information maximization promotes a representation space where classes are well-separated and individually compact, which aligns naturally with the properties encouraged by the Staircase activation functions.

## K   APPENDIX OF PROOF OF THEOREM 1

*Proof.*

$$L_{mae} = \frac{1}{N} \sum_{n}^{N} |S(z_n) - y_n|$$

Since $S(z_{c_k}) = y_n$

$$\because \textit{In interval } k, \; |S(z_n) - S(z_{c_k})| \leq 50K \times |z_n - z_{c_k}|$$
$$\therefore \; \textit{minimize } \frac{1}{N} \sum |z_n - z_{c_k}|$$
$$\equiv \; \textit{minimize } \frac{1}{N} \sum ||S(z_n) - S(z_{c_k})|$$
$$\equiv \; \textit{minimize } L_{mae}$$

That is, *minimize* $L_{mae} \equiv$ *minimize* $\frac{1}{N} \sum |z_n - z_{c_k}|$ ☐

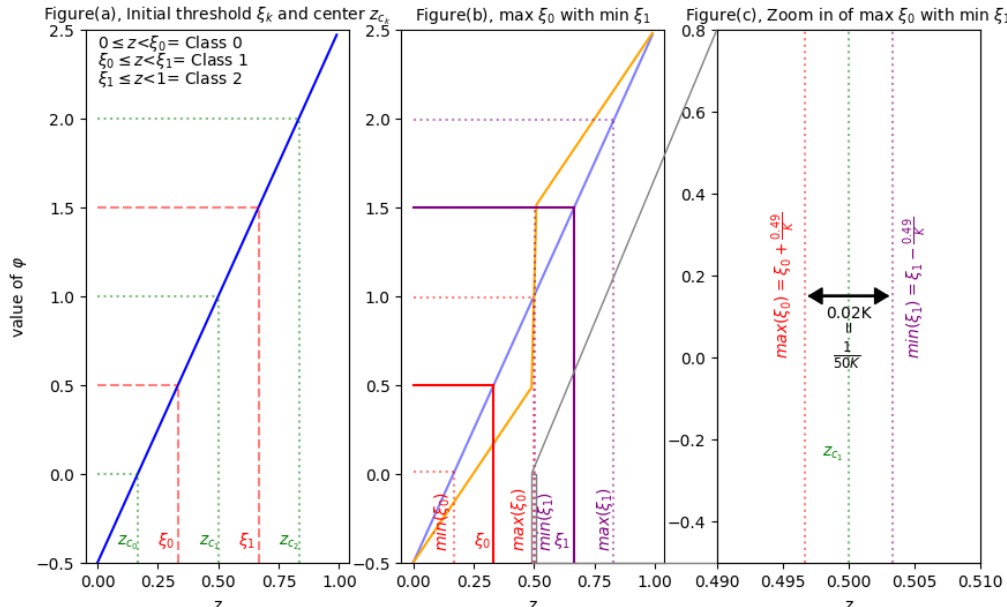

Figure 4: Figure (a) illustrates the initial state of the Piecewise Staircase activation function, where the parameters $\xi_k$ are initialized to $\frac{1}{K}$. Consequently, the initial state of the Piecewise Staircase activation function forms a linear function, with $z_{c_i}$ representing the center of the interval for each class. In Figure (b), we observe the adjustments of $\xi_0$ to maximize and $\xi_1$ to minimize during the training process. As a result, the interval of class 1 becomes the smallest, with the distance of $z$ for class 1 being $\frac{1}{50}K$. The Piecewise Staircase activation function transitions from its initial state (depicted by the blue solid line) to the new state (represented by the orange solid line). Figure (c) illustrates the zoom in version of the interval of class 1.

## L  APPENDIX OF DETAILED PROOFS OF ANALYSIS WITH MUTUAL INFORMATION

### L.1  SETTINGS

For notations and some other settings:

- $\Xi := \{\xi_1 < \xi_2 < \cdots < \xi_{K-1}\}$
- $S_\Xi : \mathcal{Z} \to \mathbb{R}$
- for the PW activation, if $z_n \in \mathcal{Z}_k$ then $S_\Xi(z_n) \to (k - 0.5, k + 0.5)$
- $\mathcal{Y} = \{1, 2, ..., K\}$
- $\mathcal{Z} = \bigcup_{k=1}^{K} \mathcal{Z}_k$
- $\mathcal{Z}_k \subseteq [\xi_{k-1}, \xi_k)$
- without loss of generality, by normalizing it is natural to assume $\mathcal{Z} \subseteq [0, 1]$
  i.e., $[0, 1] \subset [\xi_0, \xi_1) \cup [\xi_1, \xi_2) \cup \cdots \cup [\xi_{K-1}, \xi_K]$ by letting $\xi_0 := 0$, and $\xi_K := 1$.
- during the training process, denote $z_n(t) = z_{k,n}(t)$ to indicate that $z_n(t)$ is classified into the slot $[\xi_{k-1}, \xi_k)$ as the $n$-th sample of the timestamp $t$.

For a fixed $k \in [K]$ and time $t$, denote:

- $z_{k,min}(t) := \min\{z_{k,n}(t)\}$
- $z_{k,max}(t) := \max\{z_{k,n}(t)\}$
- $c_k := \frac{1}{|\mathcal{Z}_k(t)|} \sum_{n=1}^{|\mathcal{Z}_k(t)|} z_{k,n}(t)$, i.e. the center of $\mathcal{Z}_k(t)$.

Denote $y_{z_{k,n}(t)}$ as the ground truth corresponding to the sample $z_{k,n}(t)$, i.e., the $z_n(t)$ is classified correctly if and only if $y_{z_{k,n}(t)} = k$. In other words, an incorrectly classified $z_{k,n}(t)$ means the truth label is not equal to the index of the located slot, i.e., $y_{z_{k,n}(t)} \neq k$.

Note that

(1) $y_{z_{k,n}(t)} = k$ implies $|z_{k,n}(t) - c_k| = |z_{k,n}(t) - c_{y_{z_{k,n}}}|$, and

(2) $y_{z_{k,n}(t)} \neq k$ implies $|z_{k,n}(t) - c_k| < |z_{k,n}(t) - c_{y_{z_{k,n}}}|$.

$$R_{egularizer} := -\sum_{k=1}^{K} \left( |z_{k,min} - z_{k-1,max}| \right)$$

**Assumption A.1**: Training with the loss function defined by $MAE + R_{egularizer}$ converges.

**Assumption A.2**: There are at least two different samples belong to any class.

It is well-accepted that the higher the mutual information between the output variable ($Y$; the label) and the hidden variable $Z$, i.e., $\mathcal{I}(Z, Y)$, the better a deep-learning model is trained.

Denote "$A$ is equivalent to $B$" as "$A \equiv B$". We claim that minimizing the training loss function is equivalent to maximizing the mutual information.

**Lemma 6.** *If Assumption A.1 holds, or alternatively $(Z^c|Y) \sim Laplace(z_{c_k}, I)$, then*

$$\min\{H(Z|Y)\} \equiv \min\{MAE\}$$

**Lemma 7.** *If Assumption A.2 holds, then*

$$\min\{-H(Z)\} \equiv \min\{Regularizer\}$$

**Theorem 2.** *If assumptions hold, then*

$$\min\{MAE + R_{egularizer}\} \equiv \max\{\mathcal{I}(Z, Y)\}.$$

## L.2 PROOF ROUTE:

- Training well $\implies$ larger $\mathcal{I}(Z, Y)$.
- $\mathcal{I}(Z, Y) = H(Z) - H(Z|Y)$

Step1 :
$$\min \{MAE + R_{egularizer}\}$$
$$\equiv \max \{\mathcal{I}(Z, Y)\}$$
$$\equiv \min \{-\mathcal{I}(Z, Y)\}$$
$$\equiv \min \{-H(Z) + H(Z|Y)\}$$
$$\equiv \min \{H(Z|Y) - H(Z)\}$$

- Claim1:
$$\min \{H(Z|Y)\} \equiv \min \{MAE\}$$

- Claim2:
$$\min \{-H(Z)\} \equiv \min \{Regularizer\}$$

Step2 (Pf idea for Claim1)

- Claim1.1:
$$\min \{H(Z|Y)\} \equiv \min \{CE(Z; Z^C|Y)\}$$

pf key:

$$CE(Z; Z^C|Y) = H(Z|Y) + D_{KL}(Z||Z^c|Y)$$
$$i.e., H(Z|Y) = CE(Z; Z^C|Y) - D_{KL}(Z||Z^c|Y)$$

  – Claim1.1.1:
  $\min \{CE(Z; Z^c|Y)\} \equiv \min \{\sum |z_n - z_{c_k}|\}$
  pf key:

$$CE(Z; Z^c|Y) = -\mathbb{E}_{z \sim Z|Y}[log \, p_{Z^c|Y}(z)]$$
$$\overset{mc}{\approx} \frac{-1}{N} \sum_{n}^{N} log(e^{(||z_n - z_{c_k}||)}) + const$$
$$\overset{c}{=} \sum_{n=1}^{N} ||z_n - z_{c_k}||$$

  – Claim1.1.2:
  $D_{KL}(Z||Z^c|Y)$ is constant
  (when training converge of $D_{KL}(Z||Z^c|Y)$)

- Claim1.2:
$$\min \{CE(Z; Z^C|Y)\} \equiv \min \{MAE\}$$

pf key:
$MAE \leq S^* \times \sum_{n=1}^{N} ||z_n - z_{c_k}||$
$S^*$ is NOT depend on training data
$\implies \min \left\{\sum_{n=1}^{N} ||z_n - z_{c_k}||\right\} \equiv \min \{MAE\}$

Step3 (Pf idea for Claim2)
$R_{egularizer} := -\sum_{k}^{K-1} |max(\boldsymbol{z_k}) - min(\boldsymbol{z_{k+1}})|$
Note that the negative sign in the right side!

- Claim2.1:
$$\max \{H(Z)\} \equiv \min \{R_{egularizer}\}$$

## M APPENDIX OF DATASET DETAILS

Full information provided in Table. 1

Table 1: Summary of the datasets used for evaluation across five task domains. All datasets exhibit ordinal label structures reflecting severity, progression, or intensity levels.

| Task | Dataset | Data Type | Data format | Data size | Classes | Backbone | Literature method |
|------|---------|-----------|-------------|-----------|---------|----------|-------------------|
| Time Series Forecasting | AQI | Raw data | 14×18 | 43,800 | 7 | MLP | Softmax and crossentropy NNOP Cheng et al. (2008) CORAL Cao et al. (2020) |
| Age Estimate | AFAD | Image | 256×256×3 | 164,432 | 26 | ResNet34 | Multiple Output CNN Niu et al. (2016) |
| | MORPH | | | 55,134 | 55 | | CORAL Cao et al. (2020); BEL Shah et al. (2022) |
| | Abalone | Raw data | 10 | 4,177 | 10 | MLP | Unimodel Cardoso et al. (2025) |
| Malignant Tumor Detection | BUSI | Image | 128×128×3 | 780 | 3 | ResNet18 | MORF Lei et al. (2022) CORE Lei et al. (2024) |
| Monocular Depth Estimation | KITTI NYUv2 | Image | 352x1216x3 640x480x3 | 23,810 36,907 | 80 10 | GLPDepth SwinTransformerTiny | GLPDepth Kim et al. (2022) Adabins Bhat et al. (2021) OrdinalEntropy Zhang et al. (2023) |
| Sentiment Analysis | SST-5 | Sentence | 128 | 11,855 | 5 | BERT-Tiny | OLL Castagnos et al. (2022) |

## M.1 AQI DATASET

The goal of Time-series forecasting is to use observations $X^{(0:n-hour)}$ (included 14 features Temp, CO, NO, NO2, NOx, O3, PM10, PM2.5, Rainfall, RH, SO2, Wind-cos, Wind-sin) to train a model with ground truth labels $Y^{(hour:n)}$ for forecasting purposes. In this study, we have chosen 8 hours as the forecasting time horizon. We have converted the PM2.5 measurements into AQI values using the threshold table published by Taiwan's Environmental Protection Administration and used AQI as our ground truth label $Y$. We have verified our Staircase activation functions using these threshold values.

To account for the time series nature of the data, we divided the dataset into training, validation, and test sets using a 3yr / 1yr / 1yr split with sequentially, as opposed to randomly splitting the data into 5 folds.

## M.2 AFAD DATASET

The Asian Face Age Dataset (AFAD) is proposed to evaluate the performance of age estimation. This dataset is oriented to age estimation on Asian faces. Which is the largest dataset for age estimation to date. There are 164,432 well-labeled photos in the AFAD dataset. It consist of 63,680 photos for female as well as 100,752 photos for male, and the ages range from 15 to 40  Niu et al. (2016).

we divided the dataset into training, validation, and test sets using an 80% / 10% / 10% split with randomly, and generate 5-folds with a similar data distribution. Since the ages range starts from 15, hence we minus 15 for all ages, to adjust the range from 0 to 25 for training ordinal tasks.

## M.3 MORPH-ALBUM2 DATASET

The MORPH longitudinal facial recognition database is the largest longitudinal facial recognition database in the world. The longitudinal aspect of the database indicates that there are numerous images of a given subject, over time. There are 55,134 well-labeled photos in the MORPH-II dataset. It consist of 8,489 photos for female as well as 46,645 photos for male, and the ages range from 16 to 77  Ricanek & Tesafaye (2006).

We select the ages range from 16 to 70 to follow the previous work.  Cao et al. (2020); Shah et al. (2022).  we divided the dataset into training, validation, and test sets using an 80% / 10% / 10%

split with randomly, and generate 5-folds with a similar data distribution. Since the ages range starts from 16, hence we minus 16 for all ages, to adjust the range from 0 to 54 for training ordinal tasks.

### M.4 ABALONE DATASET

The Abalone10 dataset is derived from the UCI Abalone regression set by discretizing the original "Rings" values into ten ordered classes. It comprises 4,177 examples, each with 10 input features: 3 one-hot indicators for the Sex, attribute (M, F, I) and seven continuous measurements (Length, Diameter, Height, Whole weight, Shucked weight, Viscera weight, Shell weight). All features are standardized via z-normalization. The target label is obtained by binning the original ring counts into ten equally spaced intervals (zero-based indexing), yielding $K = 10$ classes. Data are split into training and test folds according to the supplied `fold` and `rep` parameters. We used the same setting and data preprocessing as Unimodal Cardoso et al. (2025).

### M.5 BUSI DATASET

BUSI dataset include breast ultrasound images among women between the ages of 25 and 75. This data was collected in 2018. The number of patients is 600 female patients. The dataset consists of 780 images with an average image size of 500x500 pixels. The images are in PNG format. The ground truth images are presented with original images. The images are classified into three classes, which are normal, benign, and malignant BUS.

We resize image from 500x500 to 128x128 to follow previous work Lei et al. (2022), and divided the dataset into training, validation, and test sets using an 80% / 10% / 10% split with randomly, and generate 5-folds with a similar distribution of data.

### M.6 KITTI DATASET

KITTI equipped a standard station wagon with two high-resolution color and grayscale video cameras. Accurate ground truth is provided by a Velodyne laser scanner and a GPS localization system. KITTI datasets are captured by driving around the mid-size city of Karlsruhe, in rural areas, and on highways. Up to 15 cars and 30 pedestrians. KITTI contains over 93 thousand depth maps with corresponding raw LiDaR scans and RGB images. Geiger et al. (2013). We used the same data preprocessing and split data partition with Kim et al. (2022).

### M.7 NYUv2 DATASET

NYUv2 dataset was captured using a Microsoft Kinect v1 sensor, providing synchronized RGB images and depth maps at a resolution of 640×480. The recordings span 464 distinct indoor scenes (living rooms, offices, kitchens, classrooms, etc.), yielding over 400,000 frames. Among these, 1,449 frames have been densely annotated with pixel-wise semantic labels across 40 categories. We employ this annotated subset for depth estimation experiments, following the standard training/testing split as defined in Silberman et al. (2012).

### M.8 SST-5 DATASET

The Stanford Sentiment Treebank is a corpus with fully labeled parse trees that allows for a complete analysis of the compositional effects of sentiment in language. The corpus is based on the dataset introduced by Pang and Lee (2005) and consists of 11,855 single sentences extracted from movie reviews. Each phrase is labeled as negative, somewhat negative, neutral, somewhat positive or positive Socher et al. (2013). We padding each sentences to 128 to follow previous work Castagnos et al. (2022), and use the same split partition of the dataset to compare the 5-folds performances.

## N  APPENDIX OF MODEL ARCHITECTURE

To assess the efficiency of our approach, we chose a neural network structure that is same to the one used in previous literature. We preferred to use the available code that can reproduce the performance of the literature. We modified the last layer by replacing it with Staircase activation functions.

(e.g., In the Monocular depth estimation task, the last layer multiplied the normalized value with the maximum depth value to recover the prediction value. We utilized Staircase activation functions to replace the operation of multiplying the maximum depth value.) Models with specially designed architectures, such as OrdinalCLIP Li et al. (2022), are not included in our comparisons, as their activation functions cannot be directly replaced. Our goal is to evaluate the effectiveness of our proposed activation function by applying it to models that rely on loss functions, label designs, or standard activations such as Sigmoid and Softmax.

## O   IMPLEMENT DETAIL

For all tasks, we use the *Adam* optimizer, and 0.0001 *learning rate* as initial value, we use *ReduceLROnPlateau* function from PyTorch to decrease the learning rate.

1. **Time Series Forecasting** :The model was trained for 200 epochs with a learning-rate patience of 10 and a batch size of 365. Training was repeated five times to compute the standard deviation.

2. **Age Estimate** : The model was trained for 50 epochs with a learning-rate patience of 5 and a batch size of 256. Training only reports the best result, the standard deviation is zero.

3. **Malignant Tumor Detection** :The model was trained for 500 epochs with a learning-rate patience of 10 and a batch size of 8. Training was repeated five times to compute the standard deviation. Training only reports the best result, the standard deviation is zero.

4. **Monocular Depth Estimation**: We adopt the same hyperparameters and backbone as the comparison methods. Specifically, we train for 30 epochs on KITTI Kim et al. (2022) and 50 to 100 epochs with early stop on NYUv2 **?**. Since the Swin-Transformer Tiny used in **?** is too large, we reduced its parameter count by a factor of six. Training only reports the best result, the standard deviation is zero.

5. **Sentiment Analysis** : The model was trained for 100 epochs with a learning-rate patience of 15 and a batch size of 200. Training was repeated five times to compute the standard deviation.

We use matrix multiplication to implement Staircase activation functions, enabling efficient computation of output formats across different dimensions. All of our tasks are running on NVIDIA A100 GPUs with 80GB memory and 1TB physical memory. The code of original paper Cao et al. (2020); Shah et al. (2022); Kim et al. (2022); Lei et al. (2022); Castagnos et al. (2022) are download from Github except MetaOrdinal. We download the BUSI BUS dataset which used in MetaOrdinal and build the same backbone (ResNet18) with Huggingface  huggingface (2022).

During the training phase, we have imposed constraints on the updating range of the thresholds in the proposed approach, with an adjustment range of $\pm0.49$ the initial value for each threshold, in order to avoid occurrences that do not conform to the desired criteria. For instance, in the case of a three-class classification task with an initial threshold of 0.333, the permissible adjustment range would be between 0.1698 and 0.4961.

# P  APPENDIX OF EXPERIMENTS RESULT IN DETAILS.

Where the prefix CE indicate cross-entropy, which denote standard classification.

Table 2: Evaluate the proposed method against the baseline on AQI dataset.

| | AQI (next 8 hour) | | | | | | | |
|---|---|---|---|---|---|---|---|---|
| Model | Precision ↑ | Recall ↑ | F1-Score ↑ | Acc ↑ | MAE ↓ | MSE ↓ | $\tau_b$ ↑ | $R_s$ ↑ |
| CE-MLP | $\mathbf{0.746}_{0.00}$ | $0.702_{0.01}$ | $0.679_{0.01}$ | $0.702_{0.01}$ | $0.306_{0.01}$ | $0.323_{0.01}$ | $0.483_{0.00}$ | $0.492_{0.00}$ |
| NNOP-MLP | $0.739_{0.00}$ | $0.707_{0.01}$ | $0.692_{0.01}$ | $0.707_{0.01}$ | $0.301_{0.01}$ | $0.317_{0.01}$ | $0.487_{0.00}$ | $0.496_{0.00}$ |
| CORAL-MLP | $0.689_{0.02}$ | $0.601_{0.00}$ | $0.464_{0.01}$ | $0.601_{0.00}$ | $0.438_{0.00}$ | $0.523_{0.01}$ | $0.129_{0.02}$ | $0.131_{0.03}$ |
| PW | $0.734_{0.00}$ | $\mathbf{0.735}_{0.00}$ | $0.718_{0.00}$ | $\mathbf{0.735}_{0.00}$ | $\mathbf{0.272}_{0.00}$ | $0.287_{0.00}$ | $\mathbf{0.518}_{0.01}$ | $\mathbf{0.527}_{0.01}$ |
| PW-MI | $0.731_{0.00}$ | $\mathbf{0.735}_{0.00}$ | $\mathbf{0.722}_{0.00}$ | $\mathbf{0.735}_{0.00}$ | $\mathbf{0.272}_{0.00}$ | $\mathbf{0.286}_{0.00}$ | $0.510_{0.01}$ | $0.519_{0.01}$ |
| AS-SoftSigmoid | $0.733_{0.00}$ | $0.721_{0.00}$ | $0.707_{0.00}$ | $0.721_{0.00}$ | $0.286_{0.00}$ | $0.301_{0.00}$ | $0.494_{0.00}$ | $0.503_{0.00}$ |
| AS-SoftSigmoid-MI | $0.735_{0.00}$ | $0.723_{0.00}$ | $0.713_{0.01}$ | $0.723_{0.00}$ | $0.284_{0.00}$ | $0.298_{0.00}$ | $0.498_{0.00}$ | $0.507_{0.00}$ |
| AS-HardSigmoid | $0.732_{0.00}$ | $0.723_{0.00}$ | $0.710_{0.00}$ | $0.723_{0.00}$ | $0.284_{0.00}$ | $0.299_{0.00}$ | $0.495_{0.00}$ | $0.504_{0.00}$ |
| AS-HardSigmoid-MI | $0.734_{0.00}$ | $0.721_{0.00}$ | $0.710_{0.01}$ | $0.721_{0.00}$ | $0.286_{0.00}$ | $0.300_{0.01}$ | $0.496_{0.00}$ | $0.505_{0.00}$ |
| NoiseSoftSigmoid | $0.731_{0.00}$ | $0.723_{0.00}$ | $0.710_{0.00}$ | $0.723_{0.00}$ | $0.284_{0.00}$ | $0.299_{0.00}$ | $0.496_{0.00}$ | $0.505_{0.00}$ |
| NoiseSoftSigmoid-MI | $0.733_{0.00}$ | $0.723_{0.00}$ | $0.710_{0.01}$ | $0.723_{0.00}$ | $0.284_{0.00}$ | $0.299_{0.00}$ | $0.501_{0.01}$ | $0.510_{0.01}$ |
| NoiseHardSigmoid | $0.732_{0.00}$ | $0.722_{0.00}$ | $0.713_{0.00}$ | $0.722_{0.00}$ | $0.284_{0.00}$ | $0.298_{0.00}$ | $0.493_{0.00}$ | $0.502_{0.00}$ |
| NoiseHardSigmoid-MI | $0.732_{0.00}$ | $0.724_{0.00}$ | $0.713_{0.00}$ | $0.724_{0.00}$ | $0.283_{0.00}$ | $0.297_{0.00}$ | $0.499_{0.01}$ | $0.508_{0.01}$ |

AS denotes Ascending. Training was repeated five times to compute the standard deviation. ↑ indicates that a higher value is better, and ↓ indicates that a lower value is better.

Table 3: Evaluate the proposed method against the baseline on AFAD and MORPH dataset.

| | AFAD | | MORPH | |
|---|---|---|---|---|
| Model | MAE ↓ | CS5 ↑ | MAE ↓ | CS5 ↑ |
| CE-CNN† | 3.60 | − | 3.34 | − |
| OR-CNN | 3.51 | 0.74 | 2.58 | 0.71 |
| CORAL-CNN† | 3.47 | − | 2.49 | − |
| BEL† | 3.13 | 0.80 | 2.33 | 0.91 |
| BEL-ResNet50 | 3.11 | 0.82 | 2.27 | $\mathbf{0.93}$ |
| PW† | $2.970_{0.00}$ | $0.849_{0.00}$ | $2.207_{0.00}$ | $\mathbf{0.929}_{0.00}$ |
| PW-MI† | $2.948_{0.00}$ | $\mathbf{0.856}_{0.00}$ | $2.217_{0.00}$ | $0.925_{0.00}$ |
| AS-SoftSigmoid† | $2.954_{0.00}$ | $0.850_{0.00}$ | $2.239_{0.00}$ | $0.927_{0.00}$ |
| AS-SoftSigmoid-MI† | $2.963_{0.00}$ | $0.852_{0.00}$ | $2.220_{0.00}$ | $0.926_{0.00}$ |
| AS-HardSigmoid† | $2.967_{0.00}$ | $0.853_{0.00}$ | $2.220_{0.00}$ | $0.927_{0.00}$ |
| AS-HardSigmoid-MI† | $2.965_{0.00}$ | $0.854_{0.00}$ | $\mathbf{2.206}_{0.00}$ | $0.928_{0.00}$ |
| NoiseSoftSigmoid† | $\mathbf{2.945}_{0.00}$ | $0.855_{0.00}$ | $2.223_{0.00}$ | $0.926_{0.00}$ |
| NoiseSoftSigmoid-MI† | $2.955_{0.00}$ | $0.853_{0.00}$ | $2.212_{0.00}$ | $0.928_{0.00}$ |
| NoiseHardSigmoid† | $\mathbf{2.945}_{0.00}$ | $0.854_{0.00}$ | $2.219_{0.00}$ | $\mathbf{0.929}_{0.00}$ |
| NoiseHardSigmoid-MI† | $2.960_{0.00}$ | $\mathbf{0.856}_{0.00}$ | $2.251_{0.00}$ | $0.923_{0.00}$ |

† denote with ResNet34 and AS denotes Ascending. Training is performed only once, and the best result across all epochs is reported, the standard deviation is zero. ↑ indicates that a higher value is better, and ↓ indicates that a lower value is better.

Table 4: Evaluate the proposed method against the baseline on Abalone10 dataset.

| | Abalone10 | | | | |
|---|---|---|---|---|---|
| Model | $Accuracy\% \uparrow$ | $MAE \downarrow$ | $QWK\% \uparrow$ | $\tau\% \uparrow$ | $ZME \to 0$ |
| WU-KLDIV | $57.9_{2.6}$ | $0.54_{0.02}$ | $62.4_{2.1}$ | $63.1_{2.5}$ | $-0.18_{0.02}$ |
| WU-Wass | $57.9_{2.4}$ | $0.53_{0.02}$ | $63.3_{1.6}$ | $63.2_{2.1}$ | $-0.17_{0.02}$ |
| PW | $59.737_{1.59}$ | $0.498_{0.03}$ | $66.749_{2.67}$ | $65.051_{1.65}$ | $-0.075_{0.01}$ |
| PW-MI | $60.048_{0.80}$ | $0.496_{0.01}$ | $66.293_{1.11}$ | $65.174_{1.14}$ | $-0.085_{0.01}$ |
| AscendingSoftSigmoid | $60.742_{1.35}$ | $0.483_{0.02}$ | $68.312_{1.89}$ | $66.495_{1.57}$ | $-0.078_{0.01}$ |
| AscendingSoftSigmoid-MI | $\mathbf{60.965}_{0.31}$ | $\mathbf{0.475}_{0.00}$ | $69.257_{0.79}$ | $67.068_{0.46}$ | $\mathbf{-0.071}_{0.01}$ |
| AscendingHardSigmoid | $59.522_{1.49}$ | $0.493_{0.02}$ | $68.165_{1.91}$ | $65.638_{1.58}$ | $-0.079_{0.01}$ |
| AscendingHardSigmoid-MI | $60.766_{0.54}$ | $0.478_{0.01}$ | $\mathbf{69.587}_{1.03}$ | $66.628_{0.86}$ | $-0.079_{0.01}$ |
| NoiseSoftSigmoid | $60.550_{1.44}$ | $0.487_{0.02}$ | $68.087_{2.10}$ | $66.098_{1.51}$ | $-0.080_{0.02}$ |
| NoiseSoftSigmoid-MI | $\mathbf{60.965}_{0.06}$ | $\mathbf{0.475}_{0.00}$ | $69.347_{0.44}$ | $\mathbf{67.127}_{0.29}$ | $-0.073_{0.01}$ |
| NoiseHardSigmoid | $59.809_{1.74}$ | $0.495_{0.03}$ | $67.802_{3.12}$ | $65.308_{2.14}$ | $-0.088_{0.02}$ |
| NoiseHardSigmoid-MI | $59.856_{1.97}$ | $0.490_{0.03}$ | $67.950_{2.79}$ | $65.835_{1.39}$ | $-0.090_{0.02}$ |

Training was repeated five times to compute the standard deviation. $\uparrow$ indicates that a higher value is better, and $\downarrow$ indicates that a lower value is better. $\to 0$ indicates closer to 0 is better.

Table 5: Evaluate the proposed method against the baseline on BUSI dataset.

| | BUSI | | | | | | | |
|---|---|---|---|---|---|---|---|---|
| Model | Precision $\uparrow$ | Recall $\uparrow$ | F1-Score $\uparrow$ | Acc $\uparrow$ | MAE $\downarrow$ | MSE $\downarrow$ | $\tau_b \uparrow$ | $R_s \uparrow$ |
| CE | 0.718 | 0.726 | 0.698 | 0.726 | $-$ | $-$ | $-$ | $-$ |
| CORE | $-$ | $-$ | $-$ | 0.820 | $-$ | $-$ | $-$ | $-$ |
| MORF | 0.816 | 0.767 | 0.775 | 0.809 | $-$ | $-$ | $-$ | $-$ |
| PW | $0.919_{0.04}$ | $0.901_{0.05}$ | $0.907_{0.04}$ | $0.913_{0.04}$ | $0.087_{0.04}$ | $0.087_{0.04}$ | $0.878_{0.06}$ | $0.889_{0.05}$ |
| PW-MI | $\mathbf{0.936}_{0.03}$ | $\mathbf{0.916}_{0.03}$ | $\mathbf{0.925}_{0.03}$ | $\mathbf{0.926}_{0.03}$ | $\mathbf{0.074}_{0.03}$ | $\mathbf{0.074}_{0.03}$ | $\mathbf{0.894}_{0.04}$ | $\mathbf{0.904}_{0.04}$ |
| AS-SoftSigmoid | $0.899_{0.01}$ | $0.855_{0.02}$ | $0.872_{0.02}$ | $0.885_{0.01}$ | $0.121_{0.01}$ | $0.131_{0.02}$ | $0.817_{0.03}$ | $0.831_{0.03}$ |
| AS-SoftSigmoid-MI | $0.936_{0.03}$ | $0.876_{0.03}$ | $0.897_{0.01}$ | $0.908_{0.01}$ | $0.092_{0.01}$ | $0.092_{0.01}$ | $0.871_{0.01}$ | $0.883_{0.01}$ |
| AS-HardSigmoid | $0.906_{0.03}$ | $0.837_{0.02}$ | $0.865_{0.02}$ | $0.879_{0.03}$ | $0.123_{0.03}$ | $0.128_{0.03}$ | $0.819_{0.03}$ | $0.834_{0.03}$ |
| AS-HardSigmoid-MI | $0.908_{0.03}$ | $0.869_{0.02}$ | $0.882_{0.03}$ | $0.891_{0.03}$ | $0.109_{0.03}$ | $0.109_{0.03}$ | $0.845_{0.04}$ | $0.859_{0.04}$ |
| NoiseSoftSigmoid | $0.881_{0.02}$ | $0.844_{0.02}$ | $0.860_{0.02}$ | $0.872_{0.01}$ | $0.128_{0.01}$ | $0.128_{0.01}$ | $0.816_{0.03}$ | $0.833_{0.03}$ |
| NoiseSoftSigmoid-MI | $0.919_{0.03}$ | $0.879_{0.02}$ | $0.895_{0.02}$ | $0.905_{0.02}$ | $0.097_{0.02}$ | $0.103_{0.02}$ | $0.857_{0.03}$ | $0.870_{0.03}$ |
| NoiseHardSigmoid | $0.916_{0.03}$ | $0.870_{0.01}$ | $0.888_{0.02}$ | $0.900_{0.01}$ | $0.103_{0.02}$ | $0.108_{0.02}$ | $0.848_{0.03}$ | $0.862_{0.04}$ |
| NoiseHardSigmoid-MI | $0.933_{0.01}$ | $0.896_{0.02}$ | $0.911_{0.01}$ | $0.910_{0.02}$ | $0.090_{0.02}$ | $0.090_{0.02}$ | $0.874_{0.02}$ | $0.886_{0.02}$ |

AS denotes Ascending. Training was repeated five times to compute the standard deviation. $\uparrow$ indicates that a higher value is better, and $\downarrow$ indicates that a lower value is better.

Table 6: Evaluate the proposed method against the baseline on KITTI dataset.

| | KITTI | | | | | |
|---|---|---|---|---|---|---|
| Model | $\delta_1 \uparrow$ | $\delta_2 \uparrow$ | $\delta_3 \uparrow$ | AbsRel $\downarrow$ | RMSE $\downarrow$ | RMSE log $\downarrow$ |
| Adabins | $0.964_{0.00}$ | $0.995_{0.00}$ | $0.999_{0.00}$ | $0.058_{0.00}$ | $2.360_{0.00}$ | $0.088_{0.00}$ |
| GLPDepth | $0.967_{0.00}$ | $\mathbf{0.996}_{0.00}$ | $\mathbf{0.999}_{0.00}$ | $0.057_{0.00}$ | $2.297_{0.00}$ | $0.086_{0.00}$ |
| PW | $0.968_{0.00}$ | $\mathbf{0.996}_{0.00}$ | $\mathbf{0.999}_{0.00}$ | $\mathbf{0.055}_{0.00}$ | $2.258_{0.00}$ | $\mathbf{0.084}_{0.00}$ |
| PW-MI | $\mathbf{0.969}_{0.00}$ | $\mathbf{0.996}_{0.00}$ | $\mathbf{0.999}_{0.00}$ | $0.057_{0.00}$ | $2.243_{0.00}$ | $\mathbf{0.084}_{0.00}$ |
| AscendingSoftSigmoid | $0.968_{0.00}$ | $\mathbf{0.996}_{0.00}$ | $\mathbf{0.999}_{0.00}$ | $0.056_{0.00}$ | $2.250_{0.00}$ | $\mathbf{0.084}_{0.00}$ |
| AscendingSoftSigmoid-MI | $0.967_{0.00}$ | $\mathbf{0.996}_{0.00}$ | $\mathbf{0.999}_{0.00}$ | $0.057_{0.00}$ | $\mathbf{2.239}_{0.00}$ | $0.085_{0.00}$ |
| AscendingHardSigmoid | $0.968_{0.00}$ | $\mathbf{0.996}_{0.00}$ | $\mathbf{0.999}_{0.00}$ | $\mathbf{0.055}_{0.00}$ | $2.255_{0.00}$ | $\mathbf{0.084}_{0.00}$ |
| AscendingHardSigmoid-MI | $\mathbf{0.969}_{0.00}$ | $\mathbf{0.996}_{0.00}$ | $\mathbf{0.999}_{0.00}$ | $\mathbf{0.055}_{0.00}$ | $2.249_{0.00}$ | $\mathbf{0.084}_{0.00}$ |
| NoiseSoftSigmoid | $0.967_{0.00}$ | $\mathbf{0.996}_{0.00}$ | $\mathbf{0.999}_{0.00}$ | $0.057_{0.00}$ | $2.273_{0.00}$ | $0.086_{0.00}$ |
| NoiseSoftSigmoid-MI | $\mathbf{0.969}_{0.00}$ | $\mathbf{0.996}_{0.00}$ | $\mathbf{0.999}_{0.00}$ | $\mathbf{0.055}_{0.00}$ | $2.254_{0.00}$ | $\mathbf{0.084}_{0.00}$ |
| NoiseHardSigmoid | $\mathbf{0.969}_{0.00}$ | $\mathbf{0.996}_{0.00}$ | $\mathbf{0.999}_{0.00}$ | $\mathbf{0.055}_{0.00}$ | $2.274_{0.00}$ | $\mathbf{0.084}_{0.00}$ |
| NoiseHardSigmoid-MI | $\mathbf{0.969}_{0.00}$ | $\mathbf{0.996}_{0.00}$ | $\mathbf{0.999}_{0.00}$ | $\mathbf{0.055}_{0.00}$ | $2.268_{0.00}$ | $\mathbf{0.084}_{0.00}$ |

Training is performed only once, and the best result across all epochs is reported, the standard deviation is zero. $\uparrow$ indicates that a higher value is better, and $\downarrow$ indicates that a lower value is better.

Table 7: Evaluate the proposed method against the baseline on NYUv2 dataset.

| NYUv2 | | | | | | |
|---|---|---|---|---|---|---|
| Model | $\delta_1\uparrow$ | $\delta_2\uparrow$ | $\delta_3\uparrow$ | AbsRel $\downarrow$ | RMSE $\downarrow$ | RMSE log $\downarrow$ |
| OrdinalEntropy | $0.537_{0.00}$ | $0.832_{0.00}$ | $0.948_{0.00}$ | $0.271_{0.00}$ | $0.849_{0.00}$ | $0.313_{0.00}$ |
| PW | $0.538_{0.00}$ | $0.834_{0.00}$ | $0.946_{0.00}$ | $0.270_{0.00}$ | $0.841_{0.00}$ | $0.312_{0.00}$ |
| PW-MI | $0.546_{0.00}$ | $\mathbf{0.844}_{0.00}$ | $\mathbf{0.953}_{0.00}$ | $0.259_{0.00}$ | $0.824_{0.00}$ | $0.303_{0.00}$ |
| AS-SoftSigmoid | $0.539_{0.00}$ | $0.835_{0.00}$ | $0.949_{0.00}$ | $0.273_{0.00}$ | $0.844_{0.00}$ | $0.309_{0.00}$ |
| AS-SoftSigmoid-MI | $\mathbf{0.560}_{0.00}$ | $0.840_{0.00}$ | $0.948_{0.00}$ | $0.271_{0.00}$ | $0.839_{0.00}$ | $0.309_{0.00}$ |
| AS-HardSigmoid | $0.537_{0.00}$ | $0.830_{0.00}$ | $0.946_{0.00}$ | $0.269_{0.00}$ | $0.849_{0.00}$ | $0.313_{0.00}$ |
| AS-HardSigmoid-MI | $\mathbf{0.560}_{0.00}$ | $\mathbf{0.844}_{0.00}$ | $0.948_{0.00}$ | $0.270_{0.00}$ | $0.823_{0.00}$ | $0.303_{0.00}$ |
| NoiseSoftSigmoid | $0.540_{0.00}$ | $0.836_{0.00}$ | $0.949_{0.00}$ | $0.269_{0.00}$ | $0.840_{0.00}$ | $0.309_{0.00}$ |
| NoiseSoftSigmoid-MI | $0.557_{0.00}$ | $\mathbf{0.844}_{0.00}$ | $\mathbf{0.953}_{0.00}$ | $\mathbf{0.253}_{0.00}$ | $0.824_{0.00}$ | $\mathbf{0.302}_{0.00}$ |
| NoiseHardSigmoid | $0.536_{0.00}$ | $0.835_{0.00}$ | $0.950_{0.00}$ | $0.268_{0.00}$ | $0.843_{0.00}$ | $0.310_{0.00}$ |
| NoiseHardSigmoid-MI | $0.554_{0.00}$ | $0.842_{0.00}$ | $\mathbf{0.954}_{0.00}$ | $0.260_{0.00}$ | $\mathbf{0.822}_{0.00}$ | $\mathbf{0.302}_{0.00}$ |

AS denotes Ascending. Training is performed only once, and the best result across all epochs is reported, the standard deviation is zero.

Table 8: Evaluate the proposed method against the baseline on SST-5 dataset.

| SST-5 | | | | | | | | |
|---|---|---|---|---|---|---|---|---|
| Model | Precision $\uparrow$ | Recall $\uparrow$ | F1-Score $\uparrow$ | Acc $\uparrow$ | MAE $\downarrow$ | MSE $\downarrow$ | $\tau_b\uparrow$ | $R_s\uparrow$ |
| CE | – | – | – | – | 0.754 | 1.171 | 0.533 | – |
| OLL | – | – | – | – | 0.739 | 1.081 | 0.544 | – |
| PW | $0.466_{0.00}$ | $0.392_{0.00}$ | $\mathbf{0.391}_{0.01}$ | $0.435_{0.00}$ | $0.709_{0.00}$ | $\mathbf{1.034}_{0.00}$ | $\mathbf{0.570}_{0.00}$ | $\mathbf{0.663}_{0.00}$ |
| PW-MI | $0.485_{0.04}$ | $\mathbf{0.393}_{0.01}$ | $0.390_{0.01}$ | $\mathbf{0.438}_{0.00}$ | $\mathbf{0.708}_{0.00}$ | $1.040_{0.01}$ | $0.566_{0.00}$ | $0.658_{0.00}$ |
| AS-SoftSigmoid | $0.466_{0.01}$ | $0.387_{0.01}$ | $0.383_{0.01}$ | $0.435_{0.01}$ | $0.717_{0.01}$ | $1.060_{0.02}$ | $0.559_{0.01}$ | $0.649_{0.01}$ |
| AS-SoftSigmoid-MI | $0.480_{0.06}$ | $0.378_{0.00}$ | $0.365_{0.01}$ | $0.432_{0.00}$ | $0.711_{0.00}$ | $1.031_{0.01}$ | $0.564_{0.00}$ | $0.652_{0.00}$ |
| AS-HardSigmoid | $0.475_{0.04}$ | $0.383_{0.01}$ | $0.379_{0.02}$ | $0.430_{0.01}$ | $0.718_{0.00}$ | $1.053_{0.01}$ | $0.561_{0.00}$ | $0.653_{0.00}$ |
| AS-HardSigmoid-MI | $0.515_{0.03}$ | $0.380_{0.01}$ | $0.369_{0.02}$ | $0.435_{0.00}$ | $0.709_{0.00}$ | $1.037_{0.01}$ | $0.564_{0.01}$ | $0.653_{0.01}$ |
| NoiseSoftSigmoid | $0.481_{0.00}$ | $0.383_{0.01}$ | $0.377_{0.01}$ | $0.434_{0.00}$ | $0.717_{0.00}$ | $1.058_{0.01}$ | $0.558_{0.00}$ | $0.648_{0.00}$ |
| NoiseSoftSigmoid-MI | $0.511_{0.03}$ | $0.374_{0.01}$ | $0.362_{0.01}$ | $0.430_{0.00}$ | $0.713_{0.00}$ | $1.036_{0.01}$ | $0.564_{0.00}$ | $0.653_{0.00}$ |
| NoiseHardSigmoid | $0.476_{0.01}$ | $0.375_{0.00}$ | $0.366_{0.01}$ | $0.427_{0.01}$ | $0.720_{0.00}$ | $1.049_{0.01}$ | $0.560_{0.00}$ | $0.650_{0.00}$ |
| NoiseHardSigmoid-MI | $0.518_{0.04}$ | $0.376_{0.01}$ | $0.366_{0.01}$ | $0.430_{0.00}$ | $0.713_{0.00}$ | $1.036_{0.01}$ | $0.563_{0.00}$ | $0.652_{0.00}$ |

AS denotes Ascending. Training was repeated five times to compute the standard deviation.

## Q  APPENDIX OF EVALUATE METRICS TABLE

Table 9: Evaluate Metrics

| Metrics | Formula |
|---|---|
| Precision $\uparrow$ | (TP)/(TP+FP) |
| Recall $\uparrow$ | (TP)/(TP+FN) |
| F1-Score $\uparrow$ | $(2 \times P \times R)/(P + R)$ |
| MAE $\downarrow$ | $1/N \sum_{n=1}^{N} |(\hat{y}_n - y_n)|$ |
| MSE $\downarrow$ | $1/N \sum_{n=1}^{N} (\hat{y}_n - y_n)^2$ |
| RMSE $\downarrow$ | $\sqrt{MSE}$ |
| RMSLE $\downarrow$ | $\sqrt{1/n \sum (log(\hat{y}) - log(y))^2}$ |
| Acc $\uparrow$ | (TP+TN)/(FP+FN+TP+TN) |
| $\delta_i \uparrow$ Kim et al. (2022) | $max\{(y/\hat{y}), (\hat{y}/y)\} < (1.25)^i$ |
| $R_s \uparrow$ Spearman (1961) | $\dfrac{(\sum(p_i-\bar{p})(q_i-\bar{q}))}{(\sqrt{(\sum(p_i-\bar{p})^2 \sum(q_i-\bar{q})^2)})}$ |
| $\tau_b \uparrow$ Kendall (1938) | $(\sum q_{ij}p_{ij})/(\sqrt{\sum q_{ij}^2 \sum p_{ij}^2})$ |
| CS $\uparrow$ Niu et al. (2016) | $N_{esi}/N \times 100\%$ |
| AbsRel $\downarrow$ Kim et al. (2022) | $1/N \sum |(\hat{y} - y)/(y)|$ |
| SilogLoss $\downarrow$ Eigen et al. (2014) | $\frac{1}{T}\sum_{i=1}^{T} d_i^2 - \frac{1}{T^2}\left(\sum_{i=1}^{T} d_i\right)^2$ |
| QWK Cohen (1968) | $1 - \frac{\sum_{i,j} w_{ij}O_{ij}}{\sum_{i,j} w_{ij}E_{ij}}$ |
| ZME | $\frac{1}{N}\sum_{k=1}^{N}\left(\hat{y}_k - y_k\right)$ |

# R   APPENDIX OF VISUALIZATION

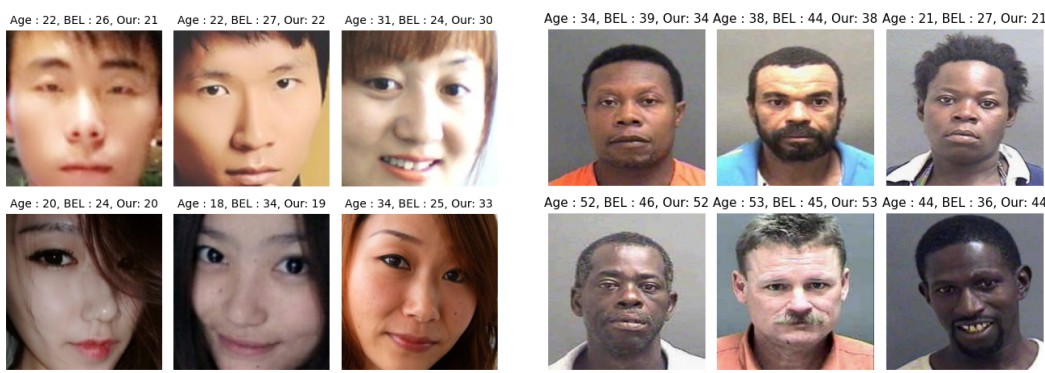

(a) AFAD dataset                    (b) MORPH2 dataset

Figure 5: Qualitative age estimation on (a) AFAD and (b) MORPH2. Each face shows "Age: *Ground Truth*, BEL: *baseline*, Our: *PW-MI*." Our predictions are consistently closer to the ground truth than the BEL baseline.

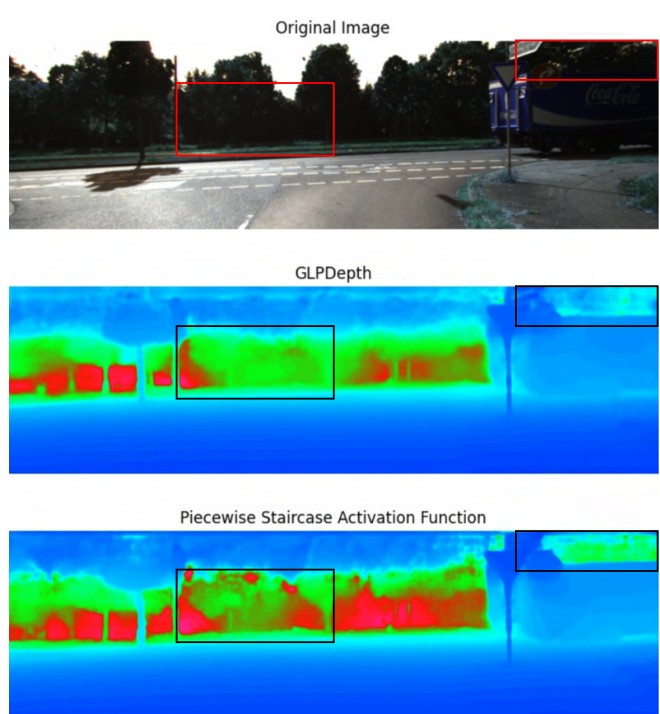

Figure 6: The trees behind the roof of the truck in the upper right red box should be positioned at a greater distance. In addition, there should be more depth between the trees in the middle red box, creating a sense of layered distance. Our proposed method is capable of distinguishing these differences on the test dataset.

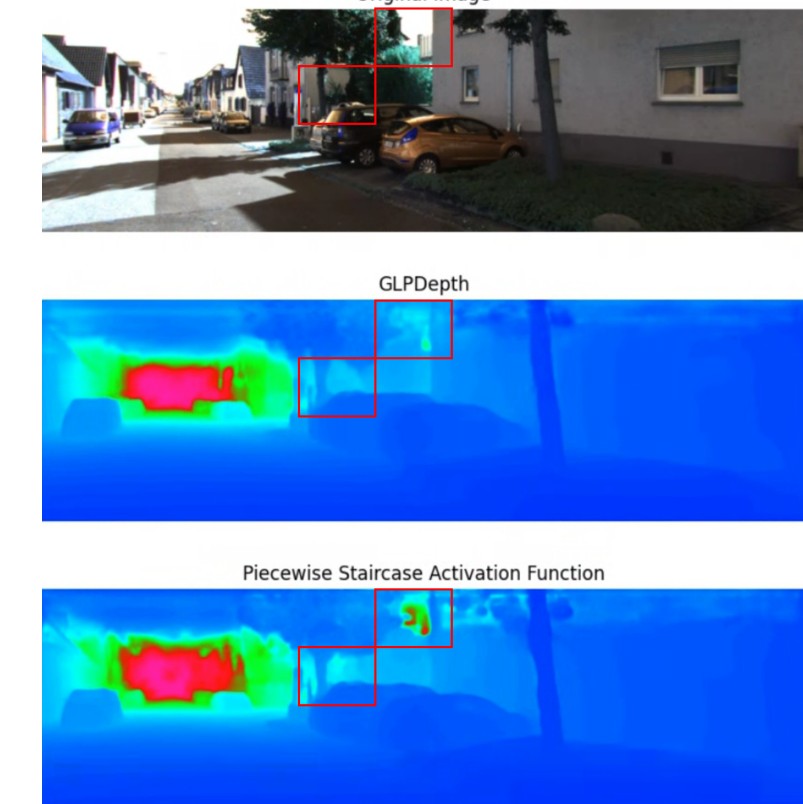

Figure 7: The red boxes highlight critical regions for qualitative analysis. In the upper-right box, the region between the tree and the house is expected to appear further in depth, a distinction more accurately captured by our method. In the middle box, the depth transitions around the tree in front of the wall are smoother and more consistent, demonstrating better spatial continuity.

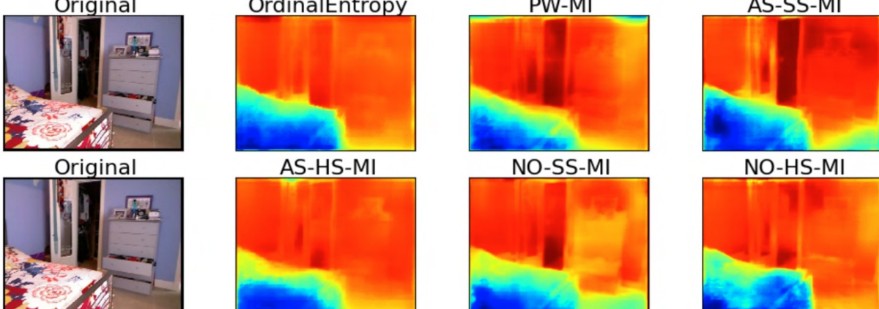

Figure 8: Visualization of depth predictions for a scene containing an open door, where the background behind the door should appear farther away (indicated by deeper red tones). AS-HS-MI correctly captures both the opening structure and the increased depth behind the door with clear spatial separation. PW-MI and AS-SS-MI also successfully predict the deeper background, although with slightly less precise boundary delineation. In contrast, NO-HS-MI identifies the door opening but underestimates the background depth.

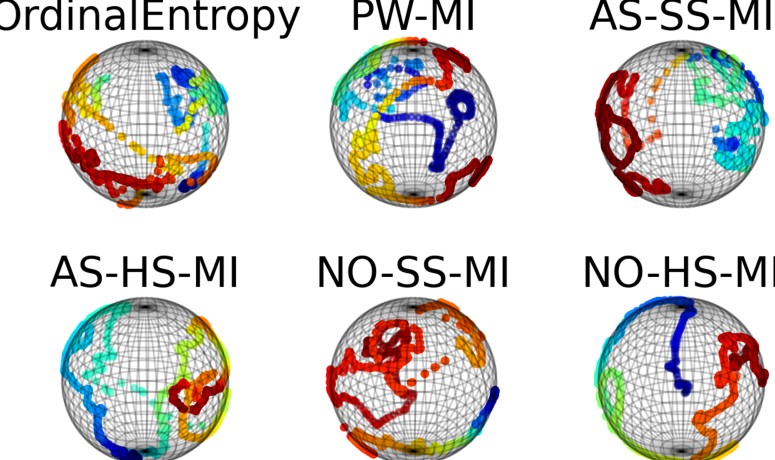

Figure 9: Latent embeddings (480×640×32) of a single NYUv2 test image are projected onto the unit sphere using t-SNE (480×640×3), visualized for six different methods. Each point represents a single pixel, and the color indicates the predicted depth value (480×640×1). We flatten the 480×640 spatial dimensions into 307,200 points, mask out the padding regions, and randomly sample 1,000 points for visualization on the sphere, though more points can be included if desired. The spatial distribution and smoothness of the trajectories vary across methods, reflecting differences in how each model organizes depth information in the latent space.

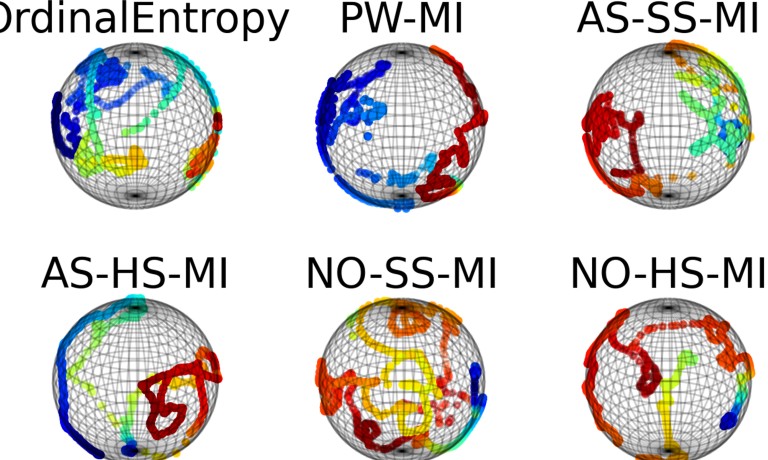

Figure 10: Latent embeddings (480×640×32) of a single NYUv2 test image are projected onto the unit sphere using t-SNE (480×640×3), visualized for six different methods. Each point represents a single pixel, and the color indicates the predicted depth value (480×640×1). We flatten the 480×640 spatial dimensions into 307,200 points, mask out the padding regions, and randomly sample 1,000 points for visualization on the sphere, though more points can be included if desired. The spatial distribution and smoothness of the trajectories vary across methods, reflecting differences in how each model organizes depth information in the latent space.

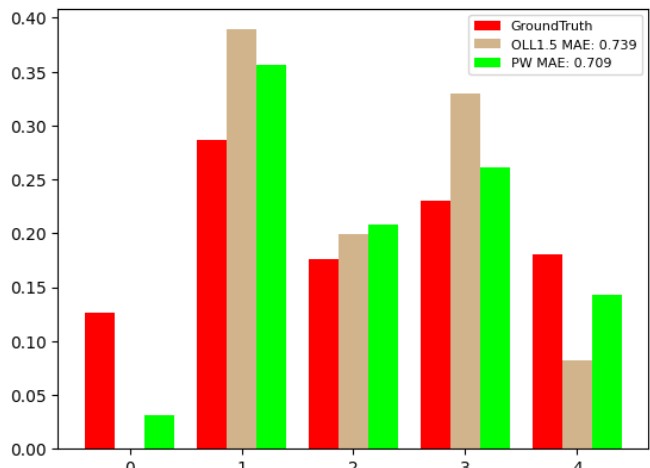

Figure 11: The histogram analysis of prediction on test dataset, where x-axis and y-axis denote label class and density, respectively.

## S  TIME COMPLEXITY

We evaluate the inference efficiency of various activation functions on the AFAD dataset under different batch sizes (1, 10, and 50). All experiments are conducted on a fixed GPU-accelerated hardware environment. The selected batch sizes represent typical deployment scenarios with varying memory and throughput constraints.

Table 10 presents the average inference time per sample (in seconds) for each activation function.

BEL Shah et al. (2022) and SlopeSoftSigmoid consistently exhibit low latency and strong scalability across batch sizes, making them particularly suitable for real-time deployment.In contrast, piecewise (PW) and noise-based variants incur increased inference time as batch size grows, reflecting their higher computational complexity.

Overall, while BEL Shah et al. (2022) achieves the lowest inference time across all configurations, the proposed activation functions maintain competitive runtime performance while delivering substantial accuracy gains.The additional computational overhead, especially for Slope-, Noise-, and Piecewise-based formulations, highlights a practical trade-off between latency and prediction quality, which is crucial for deployment in latency-sensitive applications.

Table 10: Average inference time (seconds/sample) on AFAD under varying batch sizes.

| **Activation** | **BS=1** | **BS=10** | **BS=50** |
|---|---|---|---|
| BEL Shah et al. (2022) | 0.00567 | **0.00562** | **0.00558** |
| PW | 0.00765 | 0.00869 | 0.01324 |
| SlopeSoftSigmoid | **0.00562** | 0.00633 | 0.00679 |
| SlopeHardSigmoid | 0.00825 | 0.00878 | 0.01210 |
| NoiseSoftSigmoid | 0.00957 | 0.01015 | 0.01097 |
| NoiseHardSigmoid | 0.01207 | 0.01261 | 0.01618 |