# OpenReview forum: "Data-driven Staircase Activation Functions for Ordinal Classification"
_ICLR.cc/2026/Conference — ICLR 2026 Conference Withdrawn Submission_

### Official Review · Reviewer_deNA · 2025-10-24

**Soundness:** 3
**Presentation:** 3
**Contribution:** 2
**Rating:** 4
**Confidence:** 3

**Summary:**

The paper introduces trainable staircase activation functions as a drop-in replacement for standard output layers. These activations partition the output space into learnable, ordered intervals, aligning predictions with ordinal labels.
The contribution is not so much in the core of the activation function (already proposed in the literature) but in the techniques to address the challenges that arise during training.

**Strengths:**

Well-written, addressing an important problem.

The proposed technique allows a simple drop-in Replacement for Output Layers.

The work studies three complementary remedies for training stability.

**Weaknesses:**

The citation format seems to be wrong, mixing textual with parenthetical citations (citet vc citep in natbib)

PW is used before being defined.

section 2.2
"Then f should satisfy two properties: (i) monotonicity to preserve order,
∀ xi < xj : f(xi) ≤ f(xj); and (ii) within-class closeness versus across-class separation"

1st Property: Since x_i is multidimensional, what order relationship is defined in R^d ?

2nd property: does not seem specific to ordinal regression.

Section 4.
"We evaluate it by replacing only the activation in public baselines while keeping the backbone, training protocol, and hyperparameters fixed, thus isolating its effect for fair comparison."
Being fair does not mean to do the same for everyone, but to do the best for each one. Keeping the hyperparameters fixed may favour one of the methods over the others. It would be advisable to optimize for each algorithm.

It seems that the mutual information version of the method is always the preferred one. That being the case, a bit more emphasis could have been given to its description in the main body of the document.

The methods selected in the empirical study vary from dataset to dataset, which makes the comparison harder and may give the impression that datasets/ methods were cherry-picked.

Additionally, the basic architectures trained with cross-entropy are often a strong contender, often beating the more complex alternatives; it should be included in the comparison.

**Questions:**

- see weaknesses
-How does the introduced dispersion regularizer solve the zero-gradient issue?

---

### Official Review · Reviewer_U8Lf · 2025-10-27

**Soundness:** 2
**Presentation:** 2
**Contribution:** 3
**Rating:** 4
**Confidence:** 3

**Summary:**

The paper introduces *Staircase Activation Functions* designed to preserve ordinal relationships in ordinal classification tasks. The authors identify two critical challenges during training: Degeneration, Saturation. The paper provide a theoretical analysis of these issues and propose three complementary remedies: Noise-injected Staircase, Ascending Staircase, and Piecewise-Linear Staircase. Additionally, they incorporate a Mutual Information–regularized MAE loss (LMI) to further enlarge inter-class separation and enhance ordinal structure.

**Strengths:**

- The paper is well-motivated and easy-to-follow.
- The paper introduces a novel approach that directly incorporates ordinal structure at the activation level.
- The paper provides theoretical justification via saturation/degeneration analysis and MI-based objective formulation.
- The paper demonstrates improvement across diverse modalities.

**Weaknesses:**

- The theoretical section, while insightful, is not fully rigorous. Some derivations (e.g., Proposition 1 and Lemma 1) rely on informal assumptions and omit key intermediate steps, making parts of the argument more illustrative than formally proven. The intuition is sound, but it might help if the authors explicitly frame these as heuristic analyses rather than formal proofs.

- The experiments, although showing consistent gains, do not always include the latest or strongest order-aware baselines for each domain. Including such comparisons would better contextualize the contribution and strengthen the empirical claims.

**Questions:**

1. W1

2. W2

3. Does the proposed Staircase activation risk collapsing under highly imbalanced class distributions?

4. How sensitive is performance to the threshold initialization strategy?

5. How should the $λ_{Div}$ for the MI regularizer be selected, and how sensitive is tuning?

6. Could the authors clarify the criteria used to select which Staircase variant is reported in the main tables?
Since different variants show mixed performance in the appendix, a consistent reporting rule would help address potential reproducibility or fairness concerns.

---

### Official Review · Reviewer_e6WF · 2025-10-30

**Soundness:** 2
**Presentation:** 2
**Contribution:** 2
**Rating:** 4
**Confidence:** 4

**Summary:**

this work deals with ordinal classification (OC) task.
authors propose a learnable staircase activation that can be plugged into existing models.
they also showed its saturation issue and propose different ways to alleviate that.
results are reported on 5 applications and 8 datasets in comparison to some methods.

**Strengths:**

- the writing is good.
- the work tackles an important task which is ordinal classification.
- they proposed a learnable staircase activation that can be plugged in different models.
- results are reported on different applications.

**Weaknesses:**

- writing: introduction section is very short, does not highlight the important of application of ordinal classification. in addition, it does not cover well existing works, their limitations, and why staircase activation is relevant and why this specific approach is better. background section is very long. the proposal section is more like a list of things that authors consider to remedy the saturation problem.

- poor alignment with existing works: this work is poorly aligned with existing works about OC. it is not clear why staircase activation - among all existing works- is relevant. what are existing gaps in previous works is not clear. the work seems an outlier picked - related to works mentioned in l61-, then authors attempted to reduce the saturation problem.

- results: there is a clear lack of comparison to existing works. authors discussed several works in OC in related work section. but, there is a minimal comparison to them in terms of results. it is not clear how the proposed method performs compared to them, especially recent works / SOTA.

**Questions:**

- please try to re-write the introduction, and position well the proposed method. try to justify why this approach is better than existing works - or why it worth pursuing.

**Details Of Ethics Concerns:**

none.

---

### Official Review · Reviewer_HyXS · 2025-11-03

**Soundness:** 2
**Presentation:** 3
**Contribution:** 3
**Rating:** 6
**Confidence:** 4

**Summary:**

The authors proposed data-driven staircase activation functions as drop-in loss replacement to improve deep learning for ordinal classification by correctly handling ordered categories like ratings or age groups. The main idea is a trainable staircase activation function that creates learnable, ordered output intervals in the final layer, directly matching the ordinal labels.
Direct parameterization on the formulation faces the degeneration-saturation dilemma, where gradients vanish and the function stops learning. The authors then proposed three main fixes: (i) stochastic noise injection to desaturate plateaus, (ii) a monotonic ascending term to enforce order, and (iii) adaptive piecewise-linear functions that adjust thresholds end-to-end. The authors then demonstrate performance benefits across multiple tasks: time-series, age estimation, and sentiment analysis.

**Strengths:**

There are a few things I like about the paper:
1. The paper addresses an interesting challenge in developing a proper training objective for ordinal classification problems.
2. The proposed methods can serve as a drop-in replacement for training loss, so it is applicable to any deep learning model without architecture change.
3. The proposed methods are well motivated from analyzing the degeneration-saturation dilemma.
4. The authors provide extensive theoretical analysis of the property of the proposed method.
5. The experiments are done in a range of datasets, where it shows consistent improvement.

**Weaknesses:**

1. [Experiment Baselines]. The authors only compared the proposed method with a few baselines. Adding more baselines to the experiments is suggested.
2. [Ablation]. The authors described 4 components of the proposed model. Based on the experimental results, it is not clear which components contribute the most to the performance improvement.
3. [Size of dataset]. Many of the datasets used in the experiments are relatively small.
4. [Experiment]. Some of the performance improvements in some dataset are relatively small.
5. [Experiment run]. Many experiments have 0 standard deviation, where the experiment is only done in a single run. Running experiments multiple times is always suggested.
6. [Presentation]. I suggest the author put the original metric number rather than percentage improvements. Putting percentage improvement may make it look bigger than what they are.

**Questions:**

Please address the weaknesses mentioned above.

---

### Note · Authors · 2025-11-13

I have read and agree with the venue's withdrawal policy on behalf of myself and my co-authors.